# Combining p53 mRNA nanotherapy with immune checkpoint blockade reprograms the immune microenvironment for effective cancer therapy

Yuling Xiao[1,6], Jiang Chen [2,3,6], Hui Zhou [1,4], Xiaodong Zeng [1,4], Zhiping Ruan[2,5], Zhangya Pu[2], Xingya Jiang[1], Aya Matsui[2], Lingling Zhu [2], Zohreh Amoozgar[2], Dean Shuailin Chen[1], Xiangfei Han[1], Dan G. Duda [2,7 ✉] & Jinjun Shi [1,7 ✉]

Immunotherapy with immune checkpoint blockade (ICB) has shown limited benefits in hepatocellular carcinoma (HCC) and other cancers, mediated in part by the immunosuppressive tumor microenvironment (TME). As *p53* loss of function may play a role in immunosuppression, we herein examine the effects of restoring p53 expression on the immune TME and ICB efficacy. We develop and optimize a CXCR4-targeted mRNA nanoparticle platform to effectively induce p53 expression in HCC models. Using *p53*-null orthotopic and ectopic models of murine HCC, we find that combining CXCR4-targeted p53 mRNA nanoparticles with anti-PD-1 therapy effectively induces global reprogramming of cellular and molecular components of the immune TME. This effect results in improved antitumor effects compared to anti-PD-1 therapy or therapeutic p53 expression alone. Thus, our findings demonstrate the reversal of immunosuppression in HCC by a p53 mRNA nanomedicine when combined with ICB and support the implementation of this strategy for cancer treatment.

[1] Center for Nanomedicine and Department of Anesthesiology, Brigham and Women's Hospital, Harvard Medical School, Boston, MA, USA. [2] Steele Laboratories for Tumor Biology, Department of Radiation Oncology, Massachusetts General Hospital and Harvard Medical School, Boston, MA, USA. [3] Department of General Surgery, Sir Run-Run Shaw Hospital, Zhejiang University, Hangzhou, China. [4] State Key Laboratory of Virology, Key Laboratory of Combinatorial Biosynthesis and Drug Discovery (MOE), Hubei Province Engineering and Technology Research Center for Fluorinated Pharmaceuticals, Wuhan University School of Pharmaceutical Sciences, 430071 Wuhan, China. [5] Department of Medical Oncology, First Affiliated Hospital of Xi'an Jiaotong University, Xi'an, China. [6] These authors contributed equally: Yuling Xiao, Jiang Chen. [7] These authors jointly supervised this work: Dan G. Duda, Jinjun Shi. ✉email: duda@steele.mgh.harvard.edu; jshi@bwh.harvard.edu

Loss of function in tumor suppressors is a driving force in tumorigenesis and the development of therapeutic resistance. The *p53* tumor suppressor gene, a master regulator of cell cycle arrest, apoptosis, senescence, and other cellular pathways[1], is frequently mutated in a myriad of human cancers, including hepatocellular carcinoma (HCC). Beyond cell autonomous tumor-suppressive effects, increasing evidence indicates that p53 protein can also regulate the immune tumor microenvironment (TME) by modulating interactions of tumor cells with immune cells[2–6]. For example, p53 has been shown to induce antitumor immune response via transcriptional regulation of genes encoding for key cytokines (e.g., TNF-α, IL-12, and IL-15)[7–9], chemokines (e.g., CCL2, –20, and –28, and CXCL1, –2, –3, –5, and –8)[10,11] and pathogen recognition (e.g., Toll-like receptors, TLRs)[12,13], all of which result in recruitment and activation of immune cells. Genetic restoration of p53 could induce the activation of myeloid cells to promote tumor antigen-specific adaptive immunity[14] and upregulate the NKG2D ligands on senescent tumor cells for activation of natural killer (NK) cells[15]. p53 may also play an important role in the suppression of pro-tumorigenic M2-type tumor-associated macrophage (TAM) polarization, thus facilitating antitumor immunity[16,17]. Moreover, recent studies suggest that immunogenic cancer cell death induced by cytotoxic agents may be associated with activation of the p53 pathway[18,19]. Despite these advances in understanding the role of p53, developing therapeutic approaches that directly and effectively address the loss of p53 function and its role in immunosuppression and immunotherapy resistance in HCC remains an elusive goal.

HCC is the most prevalent liver cancer with a high mortality rate and dismal prognosis[20–22]. Enhancing anti-tumor immunity using immune checkpoint blockade (ICB), including anti-CTLA-4, anti-PD-1 (aPD1), and anti-PD-L1 (aPD-L1) antibodies, has demonstrated the potential to transform the therapeutic landscape of many cancers including HCC. However, responses are seen only in a limited fraction of patients, and majority of cancer patients do not benefit from the treatment. This may be mediated in part by insufficient tumor immunogenicity and the immunosuppressive TME. Different strategies are actively being developed to improve ICB therapy in HCC, with a major focus on combining ICB with other existing therapies (such as anti-VEGF therapy), which could significantly increase anti-tumor immunity. Such combinations have been shown to improve anti-tumor efficacy in animal models and increase the survival of patients in clinical trials[23–26]. However, an increasing majority of HCC patients show no responses, and thus, new combinatorial strategies are still desperately needed.

In this work, we address the unmet need to implement p53 therapy and potentiate ICB response in HCC. We report a targeted mRNA nanoparticle (NP) platform designed to induce p53 expression and reprogram the TME, which we test in proof-of-concept studies in combination with ICB in *p53*-null murine HCC models. We optimize the p53 mRNA NP platform for HCC targeting, evaluate its therapeutic efficacy in *p53*-null HCCs growing in orthotopic and ectopic sites (alone or with aPD1 antibody), and study changes in the TME. This unique combinatorial strategy safely and effectively inhibits tumor growth in vivo, while prolonging survival and reducing ascites and metastases. Thus, combining p53 mRNA nanotherapy with ICB immunotherapy could become a transformative approach for the treatment of HCC and potentially other cancers involving p53 deficiency.

## Results

### Engineering and optimization of CXCR4-targeted mRNA NPs.
We previously developed a robust self-assembly strategy for formulating lipid-polymer hybrid NPs for mRNA delivery[27,28], composed of the ionizable lipid-like compound G0-C14 for mRNA complexation, a biocompatible poly(lactic-co-glycolic acid) (PLGA) polymer for forming a stable NP core to carry the G0-C14/mRNA complexes, and a lipid-poly(ethylene glycol) (lipid-PEG) layer for stability. We here engineered the hybrid NPs (Fig. 1a) for selective HCC targeting and high mRNA transfection efficiency. To improve HCC targeting, we modified the NPs with the targeting peptide CTCE-9908 (KGVSLSYRCRYSLSVGK; referred to as CTCE), which is specific to CXCR4, a chemokine receptor that is upregulated in cancer cells and is a validated selective target in HCC[29,30]. For comparison, we also prepared non-targeted NPs using a scrambled peptide (LYSVKRSGCGSRKVSYL; referred to as SCP). The CTCE or SCP peptide was first conjugated to 1,2-distearoyl-sn-glycero-3-phosphoethanolamine-N-[maleimide(polyethylene glycol)-3000] (DSPE-PEG-Mal) by the thiol-maleimide Michael addition click reaction, with a high chemical yield (≥82%). The chemical structures of DSPE-PEG-CTCE and DSPE-PEG-SCP were confirmed by [1]H-NMR analysis (Supplementary Fig. 1). To optimize the targeting efficacy of the mRNA NPs, we examined the effect of CTCE peptide surface density on the cellular uptake of RIL-175 murine HCC cells. As shown in Fig. 1b, CTCE-conjugated enhanced green fluorescent protein (EGFP) mRNA NPs (referred to herein as CTCE-EGFP NPs) showed significantly greater cellular uptake compared to non-targeting SCP EGFP mRNA NPs (referred to as SCP-EGFP NPs) due to the active targeting ability of the CTCE peptide towards HCC cells. We found that 5% or 6% CTCE peptide provided maximum cellular uptake in RIL-175 cells while maintaining NP stability. The uptake of the 5% CTCE-EGFP NPs was >15-fold higher than that of the 5% SCP-EGFP NPs, which was also confirmed by confocal fluorescence microscopy in RIL-175 cells (Fig. 1c). The 5% peptide density was selected for further analyses.

To identify efficacious ionizable lipid-like materials for mRNA complexation and translation, a series of G0-Cn compounds (Supplementary Fig. 2a) was synthesized through ring opening of epoxides by generation 0 of poly(amidoamine) (PAMAM) dendrimers (Supplementary Fig. 2b) and screened for using a model luciferase-mRNA. The chemical structures of G0-Cn were confirmed by [1]H-NMR spectrum (Supplementary Fig. 3). Analysis of luciferase-mRNA NPs transfection results (Fig. 1d and Supplementary Fig. 4) showed that G0-C8 had the most effective mRNA transfection ability and was thus chosen as the ionizable lipid-like material for formulating targeted mRNA NPs for in vivo treatments. To explore the possible mechanisms behind this, we studied the mRNA encapsulation efficiency and cellular uptake of the mRNA NPs formulated with different G0-Cn. As shown in the Supplementary Table 1, G0-Cn had negligible effect on the mRNA encapsulation efficacy. However, their effect on cellular uptake seemed to play an important role for the mRNA delivery efficacy (Supplementary Fig. 5), with the G0-C8 NP showing higher cellular uptake than other G0-Cn NPs.

The hybrid CTCE-conjugated p53 mRNA NPs (referred heretofore as CTCE-p53 NPs) were ~110 nm in size as measured by dynamic light scattering (DLS), and their spherical and uniform structure was confirmed by transmission electron microscopy (TEM) imaging (Fig. 1e, f). The addition of the targeting ligand (CTCE) and the scrambled peptide (SCP) to the NP surface slightly increased the particle size as well as the zeta potential, due to the positive charges of both peptides (Fig. 1f). In addition, we characterized all the nanoformulations used in this study, including Luc mRNA NPs, GFP mRNA NPs, and p53 mRNA NPs. As shown in Supplementary Fig. 6, all the nanoformulations used in this study exhibited similar average size and zeta potential.

The organic solvent DMF (dimethylformamide) had no effect on the integrity or stability of EGFP mRNA, either as naked

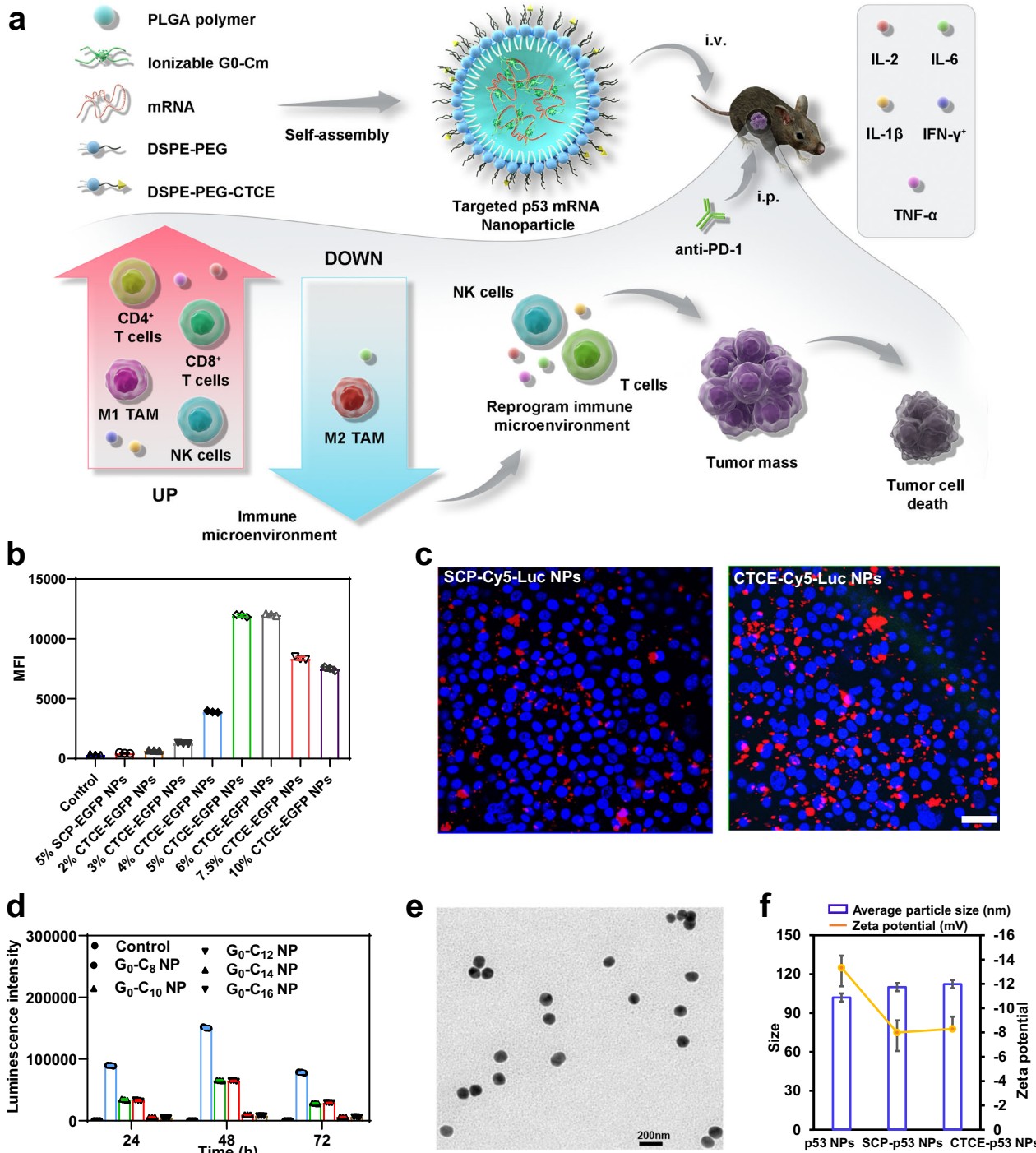

**Fig. 1 CXCR4-targeted nanoparticles (NPs) for p53 mRNA delivery to hepatocellular carcinoma (HCC). a** Schematic of CXCR4-targeted p53 mRNA NPs and combinatorial strategy using anti-PD-1 therapy to reprogram the immunosuppressive tumor microenvironment for effective treatment of p53-deficient HCC. The combination of CTCE-p53 NPs and PD-1 blockade effectively and globally reprogrammed the immune TME of HCC, as indicated by activation of CD8[+] T cells and NK cells, favorable polarization of TAMs towards the anti-tumor phenotype, and increased expression of anti-tumor cytokines. **b** Flow cytometric analysis of cellular uptake of CTCE-EGFP mRNA NPs with different CTCE peptide densities versus SCP-EGFP mRNA NPs with 5% SCP density in RIL-175 HCC cells ($n = 3$ cell samples/group). **c** Confocal fluorescence imaging of RIL-175 cell uptake of SCP-Cy5-Luciferase (Luc) mRNA NPs versus CTCE-Cy5-Luc mRNA NPs after 4 h treatment. Scale bar: 100 μm. **d** Effect of different cationic lipid-like materials G0-Cm on the transfection efficacy of Luc-mRNA NPs (mRNA concentration: 0.25 μg/mL, $n = 3$ samples/group). **e** TEM image of CTCE-mRNA NPs. Scale bar, 200 nm. **f** Average particle size and zeta potential of the p53 NPs, SCP-p53 NPs, and CTCE-p53 NPs ($n = 3$ samples/group). Data in **b**, **d**, and **f** are presented as mean values ± SD. For **c** and **e**: a representative image from one of five independent fields of view in a single experiment. Source data are provided as a Source Data file.

mRNA or encapsulated in NPs (Supplementary Fig. 7a). Moreover, we detected no obvious changes in the size of p53-mRNA NPs over a period of 96 h in the presence of 10% serum, suggesting the in vivo stability of our targeted mRNA NPs (Supplementary Fig. 7b). To further evaluate the stability of the p53-mRNA NPs, the cell viability was measured using RIL-175 cells after treatment with p53-mRNA NPs pre-incubated with 10% serum for various time points up to 96 h (at 37 °C). Comparable cell viability in all the groups (Supplementary Fig. 8) further supported the stability of these p53-mRNA NPs.

Notably, pH played a crucial role in complexing mRNA for the ionizable G0-C8, and effective mRNA complexation with G0-C8 was achieved only in acidic conditions. As shown by agarose gel electrophoresis assay at pH 7.4 (Supplementary Fig. 9), G0-C8 could not fully complex mRNA even at a weight ratio of 200 G0-C8/mRNA. In comparison, when the pH was adjusted to 3.5 in citrate buffer solution, the mRNA could be completely complexed at a weight ratio of G0-C8/mRNA as low as 2. In addition, this ratio is favorable for mRNA delivery in vivo because it reduces the need to use ionizable lipid-like materials and may thus improve the safety of the mRNA NPs. A cytotoxicity assay was further performed to evaluate the in vitro cytotoxicity of G0-C8/EGFP mRNA (Supplementary Fig. 10), which showed ~100% viability at various ratios of G0-C8/mRNA from 1 to 20 in RIL-175 cells. In addition, in vitro cytotoxicity was further examined in both RIL-175 and normal hepatocyte THLE-3 cells. The near-100% cell viability at all tested concentrations in both cell lines (Supplementary Fig. 11) indicated the safety of our mRNA NPs.

**CXCR4-targeting improves mRNA NP delivery to HCC cells in vitro and in vivo.** We then investigated the CTCE-targeting effect of our mRNA NPs on cellular uptake and mRNA transfection in p53-deficient murine HCC cells (RIL-175) using flow cytometry. We first examined the transfection efficacy of the targeted mRNA NPs and non-targeted mRNA NPs in vitro using EGFP-mRNA as the model mRNA, by counting EGFP-positive cells (Fig. 2a). Both SCP-EGFP NPs and CTCE-EGFP NPs showed markedly higher fractions (>90%) of EGFP-positive cells after mRNA NP-transfection compared to controls (free/naked EGFP mRNA). Notably, the CTCE-EGFP NPs induced a ~4.5-fold higher mean fluorescence intensity in cells compared to the SCP-EGFP NPs (Supplementary Fig. 12). The higher transfection efficiency of CTCE-EGFP NPs was confirmed by fluorescence microscopy (Fig. 2b). To further verify the selectivity of the CTCE-mRNA NPs, we also examined the targeting effect of CTCE peptide by blocking the CXCR4 receptor on RIL1-75 cell surface using free CTCE peptide (Supplementary Fig. 13). Upon treatment with CTCE peptide, the fluorescence intensity of RIL-175 cells co-incubated with CTCE-Cy5-Luciferase mRNA NPs was significantly lower than that without blocking. Moreover, we generated a CXCR4-knockout (CXCR4-KO) RIL-175 cell line by CRISPR/Cas9 editing and performed in vitro cellular uptake. As evidenced by Western blotting (WB, Supplementary Fig. 14), CXCR4 expression of the RIL-175 cells were effectively knocked out by CRISPR/Cas9 editing. In vitro cellular uptake study (Supplementary Fig. 15) showed that the fluorescence intensity of CXCR4-KO RIL-175 cells co-incubated with CTCE-Cy5-Luciferase mRNA NPs was significantly reduced than that of the sgControl RIL-175 cells (without CXCR4-KO). These results demonstrate the CXCR4-mediated active targeting effect of the CTCE-NPs on the RIL-175 cell line.

Next, intracellular uptake of the mRNA NPs in RIL-175 cells was examined by confocal fluorescence microscopy after incubating Cy5-labeled Luciferase-mRNA NPs (CTCE-Cy5-Luc NPs) with RIL-175 cells for 0.5, 2, 4, or 6 hrs. The intensity of red fluorescence from Cy5-Luc mRNA in the cells increased in proportion to incubation time (Supplementary Fig. 16), suggesting the successful intracellular delivery of our mRNA NPs.

To test the efficiency of CXCR4-mediated HCC-targeting of CTCE-mRNA NP delivery in vivo, we next conducted pharmacokinetics (PK) and biodistribution (BioD) studies. We first evaluated PK parameters by administering targeted or non-targeted Cy5-Luc-mRNA NPs or free Cy5-Luc-mRNA into healthy C57Bl/6 mice via the tail vein. The PK results showed that free mRNA was rapidly cleared, with a dramatic decrease to ~8% after 15 min (Fig. 2c). In contrast, similar to Cy5-Luc NPs without peptide modification, both SCP-Cy5-Luc NPs and CTCE-Cy5-Luc NPs showed prolonged mRNA circulation, with >30% of the Cy5-Luc-mRNA still circulating after 60 min. After 4 h, nearly 20% of both NPs were still detectable, while most free mRNA was cleared within 1 h. This result also indicated that the presence of the targeting moiety (i.e., CTCE) did not alter the PK profile of the mRNA NPs. We then evaluated the BioD and tumor accumulation of these NPs in both orthotopically and ectopically (s.c.) grafted RIL-175 HCCs. Tumor-bearing mice were administered free Cy5-Luc-mRNA, non-targeted SCP-Cy5-Luc-mRNA NPs, or targeted CTCE-Cy5-mRNA NPs by tail vein. As shown in Fig. 2d, e and Supplementary Fig. 17, in both HCC models, both NPs exhibited considerable intratumoral accumulation, while the fluorescent signal of free Cy5-mRNA was barely detectable in the tumor tissue 24 h post-injection. Notably, there was ~1.5 and 2.7-fold greater intratumoral accumulation of CTCE-targeted NPs than non-targeted NPs in the orthotopic and ectopic models, respectively. Taken together, the evidence suggests that CTCE-targeted NPs demonstrated significantly enhanced cellular uptake, mRNA transfection efficiency, and intratumoral accumulation compared to non-targeted NPs irrespective of tumor site/stroma, supporting the use of CTCE peptide ligands for selective HCC cell targeting.

**CXCR4-targeted mRNA NP increases p53 protein expression and reduces HCC cell viability in vitro.** To determine whether the targeted p53-mRNA NPs could induce the expression of therapeutic p53 in p53-null RIL-175 cells, we first checked p53 protein expression after treatment with CTCE-p53 NPs versus SCP-p53 NPs. Both WB and immunofluorescence (IF) staining (Fig. 2f, g) confirmed the successful restoration of p53 expression in RIL-175 cells. The WB data further showed that targeted NPs exhibited enhanced level of p53 expression compared with non-targeted NPs. In addition, the IF images showed that p53 protein was mainly localized in the cytoplasm of RIL-175 cells. Next, we tested cell growth and cell viability after treatment with CTCE-p53 NPs versus SCP p53 NPs. Figure 2h shows that the number of viable cells was dramatically decreased after 10-day treatment with SCP-p53 NPs or CTCE-p53 NPs compared to control-treated cells, or to cells treated with CTCE-EGFP NPs or empty CTCE-NPs. Of note, the CTCE-p53 NPs elicited greater growth inhibition than non-targeted SCP-p53 NPs, consistent with higher p53 expression. Moreover, CTCE-p53 NPs significantly decreased cell viability in a dose-dependent manner compared to the control, free mRNA, and control NPs (Fig. 2i). These results indicate that the CTCE-targeting NP system effectively delivers p53 mRNA to HCC cells, restoring functional p53 activity and reducing HCC cell viability.

In addition, we tested whether the CTCE-p53 NPs could induce the suppressing function of p53 in p53-wild type murine HCC cell line HCA-1. As shown in the Supplementary Fig. 18, modest cytotoxicity was observed at high doses in HCA-1 cells, whereas empty NPs and control NPs (CTCE-EGFP NPs) had no effects on HCA-1 cell viability.

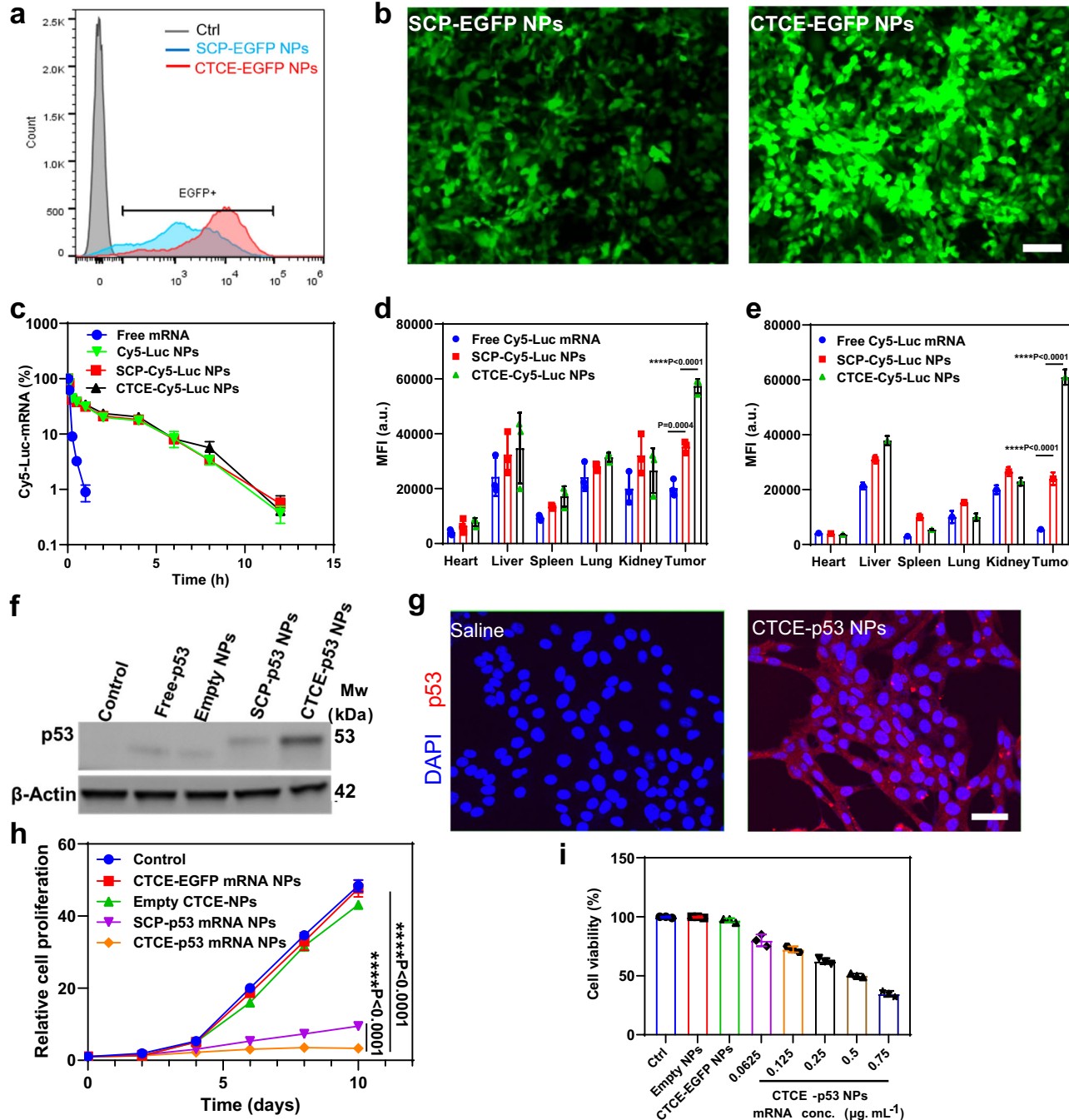

**Fig. 2 CXCR4-mediated HCC-targeting of CTCE-mRNA NPs in vitro and in vivo. a** Flow cytometry analysis of in vitro transfection efficiency (%GFP positive cells) of SCP-EGFP NPs vs. CTCE-EGFP NPs in *p53*-null RIL-175 cells. **b** Immunofluorescence of RIL-175 cells transfected with SCP-EGFP NPs vs. CTCE-EGFP NPs (magnification, ×50). Cells were treated with SCP-EGFP NPs or CTCE-EGFP NPs for 12 h and further incubated for 24 h with fresh cell culture medium (mRNA concentration: 0.5 μg/mL). Scale bar: 100 μm. **c** Circulation profile of free Cy5-Luc mRNA, SCP-Cy5-Luc NPs, and CTCE-Cy5-Luc NPs (mRNA dose: 350 μg/kg) after i.v. administration. **d**, **e** Quantification of biodistribution of free Cy5-Luciferase mRNA, SCP-Cy5-Luciferase (Luc) NPs, and CTCE-Cy5-Luc NPs in orthotopic (**d**) and ectopic (**e**) HCC grafts ($n = 3$ mice/group;) at 24 h post-i.v. injection (mRNA dose: 350 μg/kg). **f** Western blot analysis of p53 protein expression after treatments (mRNA concentration: 0.5 μg/mL). β-actin was used as the loading control. **g** Immunofluorescence for p53 in RIL-175 cells after treatment with saline or CTCE-p53 NPs (p53 mRNA concentration: 0.25 μg/mL). Scale bar: 50 μm. **h** RIL-175 cell growth rate after treatment with control (saline), CTCE-EGFP NPs, empty NPs, SCP-p53 NPs, or CTCE-p53 NPs (mRNA concentration: 0.5 μg/mL) ($n = 3$ cell samples/group). **i** RIL-175 cell viability after treatment with control (saline), empty NPs, control NPs (CTCE-EGFP NPs), or CTCE-p53 NPs with different mRNA concentrations (0.0625–0.75 μg/mL) ($n = 3$ cell samples/group). Statistical significance was calculated using one-way ANOVA with a Tukey post-hoc test. Data in **c**, **d**, **e**, **h**, and **i** are presented as mean values ± SD. **$P < 0.01$; ***$P < 0.001$; ****$P < 0.0001$. For **b** and **g**: a representative image from one of five independent fields of view in a single experiment. For **f**: this experiment was repeated five times independently with similar results. Source data are provided as a Source Data file.

**Combining CXCR4-targeted p53 mRNA NPs with PD-1 blockade inhibits tumor growth and reprograms the immune TME in orthotopic *p53*-null murine HCC.** To examine the role of p53 in immunosuppression in HCC, we tested the CTCE-p53 NPs and aPD1 against *p53*-null HCC. Mice with established orthotopic RIL-175 tumors were treated with either CTCE-p53 NPs at a mRNA dose of 350 μg/kg by intravenous (i.v.) injection, aPD1 by intraperitoneal (i.p.) injection, or their combination, every 3 days for 4 cycles (Fig. 3a). Tumor growth was monitored by high-frequency ultrasound imaging (Fig. 3b). In vivo results revealed that CTCE-p53 NPs treatment or aPD1 therapy alone inhibited HCC growth compared to IgG-treated control mice, but their combination was significantly more effective than either treatment alone (individual growth curves in Fig. 3c, mean tumor volumes in Fig. 3d, and mean tumor weight in Supplementary Fig. 19a). We also performed immunohistochemistry (IHC) analysis to confirm the expression of p53 in the orthotopic tumors. As shown in Fig. 3e, p53 was expressed at the highest levels in the CXCR4-p53 NP-treated groups, confirming the successful delivery of p53 mRNA to the orthotopic tumors.

We then examined the impact of treatment on immune cell infiltration and activation in the RIL-175 tumors by flow cytometry analyses of digested HCC tissues. Compared to treatment with CTCE-EGFP NPs, CTCE-p53 NPs, or aPD1 alone, we found that the combination of CTCE-p53 NPs with aPD1 significantly increased the number of infiltrating CD8$^+$ T cells (Fig. 3f). Importantly, the fraction of activated (IFN-γ$^+$ TNF-α$^+$) CD8$^+$ T cells was significantly increased in the HCC tissue after combination therapy (Fig. 3g). In addition, the fraction of infiltrating CD4$^+$FoxP3$^-$ effector T cells (Fig. 3h), mature (KLRG1$^+$CD11b$^+$) NK cells (Fig. 3i, j), and activated (IFN-γ$^+$ and IFN-γR$^+$) NK cells (Fig. 3k, l) all increased after combined treatment with CTCE-p53 NPs and aPD1. Moreover, we found that combination therapy effectively polarized tumor-associated macrophages (TAMs) towards the M1-like phenotype and decreased M2-like TAMs in HCC (Fig. 3m, n). It is worth noting that CTCE-p53 NPs alone increased the fractions of mature NK cells and M1 TAMs while reducing M2 TAMs (Fig. 3l–n); in contrast, aPD1 alone had the opposite effect by polarizing TAMs toward the M2-phenotype (Fig. 3m, n). We also examined changes in key immune cytokines post-treatment by multiplexed array analysis of whole tumor tissue protein extract. We found that CTCE-p53 NPs and aPD1 significantly increased TNF-α and IL-1β levels; they also tended to increase IFN-γ$^+$ and IL-2 and decrease IL-6 but neither IL-10 nor MCP1 (CCL2) (Fig. 3o–q and Supplementary Figs. 19b–d). Collectively, these results suggest that the combination of CTCE-p53 NPs and PD-1 blockade effectively and globally reprogrammed the immune TME of HCC by increasing effector immune cells and cytokine levels in the tumor.

We further compared side-by-side the survival benefit of the combination of CTCE-p53 NPs with aPD1 against a regimen similar to the new standard of care in HCC patients (i.e., anti-VEGFR2 antibody+ aPD-L1 antibody) in the orthotopic RIL-175 tumor model (Supplementary Fig. 20). Results showed that both treatments were effective and comparable in increasing overall survival and delaying disease morbidity in the *p53*-null murine HCC model. In addition, the in vivo therapeutic efficacy of the combination of CTCE-p53 NPs with aPD1 were also evaluated in an orthotopic *p53*-wild type HCC tumor model (HCA-1) in C3H mice. Though the CTCE-p53 NPs showed modest in vitro cytotoxicity in HCA-1 cells (Supplementary Fig. 18), this modest in vitro effect did not translate into an in vivo survival benefit (Supplementary Fig. 21) with the same dosage and dosing frequency used in the RIL-175 model.

**Combining CXCR4-targeted p53 mRNA NPs with PD-1 blockade is effective in ectopic *p53*-null murine HCC.** To determine whether the comprehensive reprogramming of the immune TME was dependent on the localization of tumor within the liver, we next evaluated in vivo p53 expression, anti-tumor immune response, and anti-tumor efficacy in a subcutaneously grafted HCC model in immunocompetent C57Bl/6 mice. We administered four injections of CTCE-p53 NPs i.v. (350 μg/kg body weight) and aPD1 i.p. (100 μg per dose) every 3 days in mice with established tumors (Supplementary Fig. 22a). Tumor-bearing mice treated with CTCE-EGFP NPs served as controls. We first evaluated the anti-tumor effect of CTCE-p53 NPs and aPD1 by bioluminescence imaging of the luciferase-expressing RIL-175 tumors to estimate viable tumor burdens (Fig. 4a). The combination treatment markedly limited the increase of bioluminescence signals compared to CTCE-p53 NPs or aPD1 treatment alone, indicating a potent anti-tumor effect. Moreover, RIL-175 tumor-bearing mice treated with CTCE-EGFP NPs showed aggressive tumor growth, while aPD1 treatment and CTCE-p53 NPs alone delayed the growth of RIL-175 tumors (Fig. 4b and Supplementary Fig. 22b). The combination of CTCE-p53 NPs with anti-PD1 showed a significantly greater anti-tumor effect than either treatment alone, significantly reducing tumor volume and inducing tumor regression after 4 cycles of treatment (Fig. 4b). Next, protein extracts from tumor tissues from the different treatment groups were analyzed by WB. As shown in Fig. 4c, CTCE-p53 NP treatment alone and combined with aPD1 treatment both elicited high levels of p53 protein expression in ectopic *p53*-null RIL-175 tumors, whereas neither the aPD1 nor the control NPs (i.e., CTCE-EGFP NPs) had any effect on p53 expression. IHC analysis of tumor sections further confirmed p53 expression (Supplementary Fig. 22c). These results demonstrate that the p53 mRNA NPs effectively restored p53 expression in vivo and significantly enhanced the anti-tumor effects of aPD1 therapy in HCC growing outside the liver.

Using the same model, we also harvested tumors and lymph nodes to examine the number and phenotype of immune cells and the changes in secreted cytokines after four cycles of treatment. CTCE-p53 NPs alone or in combination with aPD1 induced a significant increase in CD80$^+$CD86$^+$ lymph node-resident dendritic cells (LNDCs) and intratumoral CD8$^+$ T cells (Fig. 4d, e), and a significant decrease in M2-type TAMs (Fig. 4f). IF analysis of tumor tissues confirmed the increased intratumoral infiltration by CD8$^+$ T cells after combination treatment (Fig. 4g). Multiplexed array analysis revealed, similar to orthotopic HCCs, increased expression of cytokines associated with immune cell activation (e.g., TNF-α, IL-1β, IFN-γ, and IL-2) and also decreased expression of immunosuppressive cytokines (e.g., IL-10 and MCP-1) in the ectopic HCCs after combination treatment (Fig. 4h–k and Supplementary Fig. 23). Moreover, we also studied the role of p53 on MHC class I expression by WB and IF. Results in Supplementary Figs. 24 and 25 revealed an association between p53 and MHC class I expression, indicating the potential role of p53 restoration in inducing immune responses. These results demonstrate that targeting HCC cells with CTCE-p53 NPs combined with aPD1 therapy triggers anti-tumor immunity and reprograms the immune TME of HCC both in the liver and in other organs.

**Combination therapy prolongs survival and reduces bloody ascites, pleural effusions, and lung metastases.** Using the orthotopic RIL-175 tumor model, we further evaluated the therapeutic efficacy of combining aPD1 with CTCE-p53 NPs in mice with established tumors (Fig. 5a). We treated the mice by i.v.

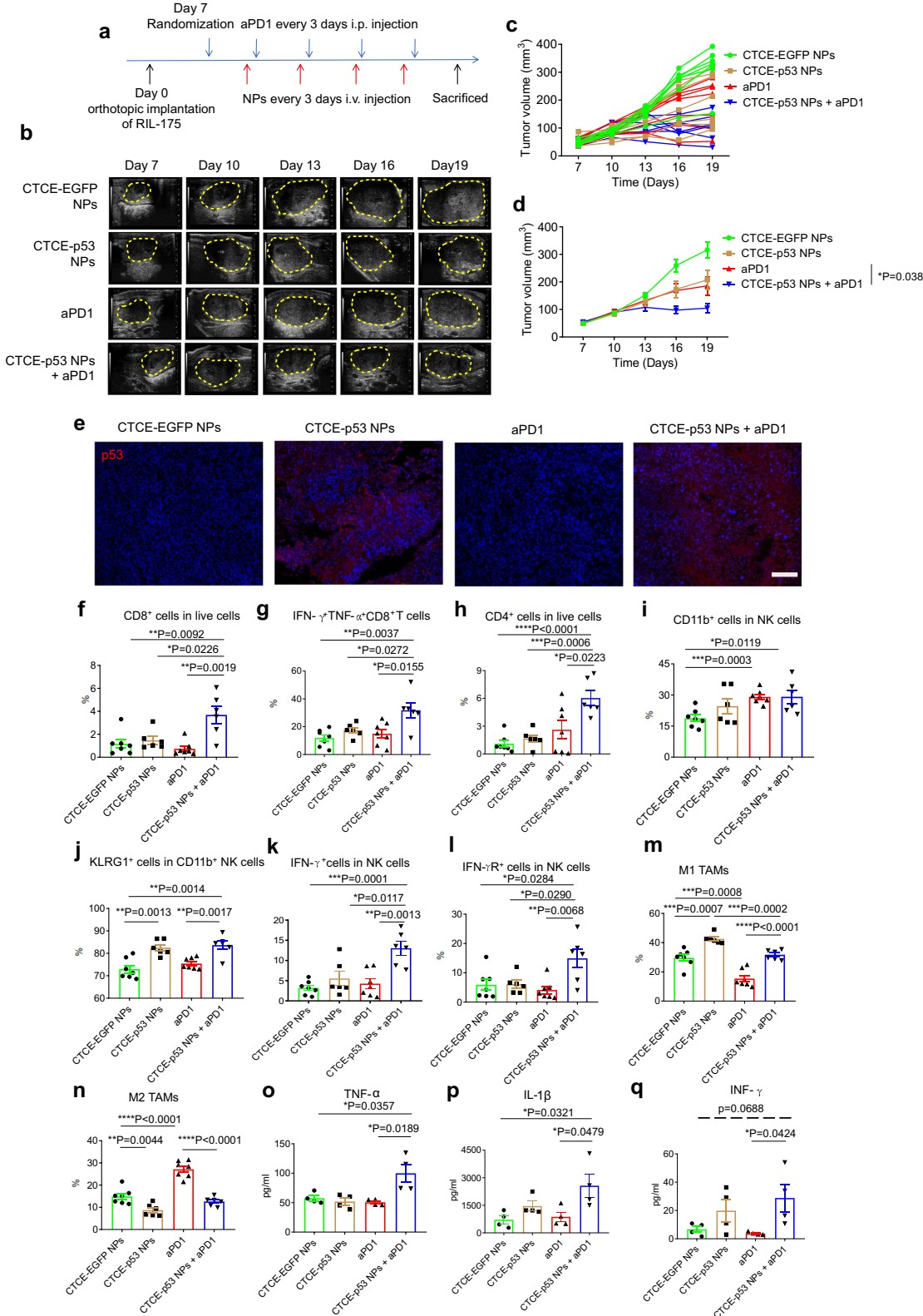

injection of NPs and i.p. injection of aPD1 for four cycles and then monitored the tumor growth by ultrasound imaging and the survival. CTCE-p53 NPs alone and aPD1 alone modestly inhibited tumor growth, but their combination elicited a significant delay in tumor growth (Fig. 5b, c). Notably, the group treated with CTCE-p53 NPs plus aPD1 showed a significant and substantial survival benefit (median overall survival of 43.5 days,

almost double that in the control group in this model, HR = 0.26; $p = 0.0001$) (Fig. 5d). In addition, only the combination treatment reduced the incidence of bloody ascites (Fig. 5e) and pleural effusions (Fig. 5f), which are potentially lethal adverse effects of orthotopic HCC. Moreover, when we assessed the lung metastatic burden by enumerating metastatic nodules, we found it significantly reduced in the group that received a combination of

**Fig. 3 PD-1 blockade combined with CXCR4-targeted p53 mRNA NPs reprograms the immune TME and promotes anti-tumor immunity in HCC. a** Timeline of tumor implantation and treatment schedule in the orthotopic HCC model. The mice with orthotopic RIL-175 tumor were treated with CTCE-EGFP mRNA NPs or CTCE-p53 mRNA NPs every 3 days for 4 i.v. injections. Anti-PD-1 (aPD1) was given at 10 mg/kg every 3 days by i.p. injection. **b** High-frequency ultrasound images of the RIL-175 orthotopic tumor-bearing C57BL/6 mice at Day 7, 10, 13, 16, and 19 ($n = 7$ mice/group). **c, d** Tumor growth profile of each indicated treatment group ($n = 7$ mice/group). **e** Immunofluorescence staining of p53 expression in RIL-175 tumors (red signals) in different groups. Scale bar: 200 μm. **f–n** Flow cytometry analysis ($n = 7$ samples for CTCE-EGFP-NPs and aPD1group; $n = 6$ samples for CTCE-p53 NPs and CTCE-p53 NPs+aPD1 group) of tumor CD8 + cytotoxic T cells (**f**), IFN-g+TNF-α+ cells among CD8+ T cells (**g**), CD4+ T cells (**h**), CD11b+ cells when gating on NK cells (**i**), KLRG1+ cells when gating on CD11b+ NK cells (**j**), IFN-g+ cells when gating on NK cells (**k**), IFN-gR+ cells when gating on NK cells (**l**), M1-like tumor-associated macrophages (TAMs) (**m**), and M2-like TAMs (**n**). **o–q** Increased levels of expression of TNF-α (**o**), IL-1β (**p**), and IFN-γ (**q**) in RIL-175 tumor tissues by protein array measurements after combination treatment ($n = 4$ tumor samples/group). Statistical significance was calculated via one-way ANOVA with a Tukey post-hoc test. All data are presented as mean ± S.E.M. For **e**: this experiment was repeated thrice independently with similar results. *$P < 0.05$; **$P < 0.01$; ***$P < 0.001$; ****$P < 0.0001$. Source data are provided as a Source Data file.

CTCE-p53 NPs with aPD1 (Fig. 5g). These findings suggest that p53 restoration using CXCR4-targeted mRNA NPs can markedly improve the efficacy of aPD1 therapy in *p53*-deficient HCC.

**Combination of p53 mRNA NPs with aPD1 is safe in vivo.** Finally, to evaluate the in vivo safety of CXCR4-targeted p53-mRNA NPs alone and in combination with aPD1, mouse weight was monitored during the above animal studies with the s.c. grafted and orthotopic models, and blood and major organs (e.g., heart, kidneys, liver, lung, and spleen) were harvested at the end of these studies. No significant change in body weight was observed in any of the treatment groups (Supplementary Figs. 26 and 27). We performed hematological analysis based on serum biochemistry and whole blood panel tests. A series of parameters were tested, including alanine aminotransferase (ALT), aspartate aminotransferase (AST), urea nitrogen (BUN), albumin, BUN, creatinine, globulin, calcium, cholesterol, phosphorus, glucose, total protein, red blood cells (RBC), white blood cells (WBC), hemoglobin (Hb), mean corpuscular hemoglobin concentration (MCHC), mean corpuscular hemoglobin (MCH), hematocrit (HCT), and lymphocytes (LY). As shown in Fig. 6 and Supplementary Fig. 28, no obvious changes were detected in any hematological parameter across groups, indicating negligible side effects of the CTCE-p53 NPs and their combination with aPD1. We also examined the major organs by H&E staining. Histological analyses revealed no obvious abnormality and no differences in the main organs among the treatment groups (Supplementary Figs. 29 and 30), further demonstrating the in vivo safety of the combination treatment.

## Discussion
The last decade has witnessed a tremendous shift in cancer treatment toward immunotherapy with ICBs, significantly extending the survival of cancer patients, including those with HCC. However, benefits are seen in only a fraction of patients. Combinations of ICB therapy with other therapy modalities (e.g., chemotherapy, radiotherapy, and targeted therapy) are being actively explored for their ability to activate anti-tumor immune response and/or alter the immunosuppressive TME. These strategies are designed to increase the recruitment of activated effector T cells in 'immunologically cold' tumors that lack T cells and do not respond to ICB-based therapy.

The tumor suppressor *p53* is one of the most frequently mutated genes in a wide range of cancers and is strongly associated with tumorigenesis, tumor progression, treatment resistance, and adverse prognosis. Compelling evidence suggests that p53 dysfunction leads to immunosuppression and immune evasion. Restoration of p53 function thus may offer the opportunity to reverse immunosuppression of the TME and improve the anti-tumor efficacy of ICB therapy. Current efforts towards p53

reactivation include small molecules and DNA therapies[31–37], which have shown notable outcomes but are also associated with formidable drawbacks[38,39], highlighting the need for new therapeutic strategies to restore p53 functions.

The use of synthetic mRNA has attracted tremendous attention, as exemplified by the recent clinical approval of COVID-19 mRNA nano-vaccines and the clinical trials of a number of mRNA nanotherapeutics for diverse diseases including cancer[28,40–43]. As a compelling alternative to DNA, mRNA requires only cytosolic delivery for translation, thus largely avoiding host genome integration and eliciting faster and more predictable protein expression. In this study, we developed a CXCR4-targeted mRNA NP platform for effective p53 restoration and tested it in combination with aPD1 immunotherapy using *p53*-null murine HCC models. We extensively optimized the p53 mRNA NP platform by screening a series of ionizable lipid-like compounds and varying densities of CXCR4-targeting ligands for improving mRNA translation and HCC targeting in vivo. Our results demonstrate that the combination of CXCR4-targeted p53 mRNA NPs with aPD1 leads to a potent antitumor effect in intrahepatic and ectopic models of HCC with p53 loss. The combination of p53 mRNA NPs and aPD1 effectively and globally reprogrammed the immune TME by promoting MHC-I expression and anti-tumor immunity, and decreasing the expression of immunosuppressive cytokines in HCC, irrespective of organ location. These findings suggest that p53 mRNA nanotherapy could enhance the efficacy of ICB therapy, substantially improving the treatment of *p53*-deficient HCC and potentially other *p53*-deficient cancers. Further studies will be required to gain an in-depth understanding of the role of p53 in immune regulation, such as how the p53 status of cancer cells (e.g., p53 mutation) affects the immune TME and how the transfection of p53 mRNA NPs in immune cells (e.g., T cells, NK cells, and macrophages) affects their function in vivo. In addition, new combinatorial strategies between p53 targeting, ICB, with or without VEGF blockade may be required to increase durability of responses. If successfully translated, the mRNA nanotherapy-based p53 restoration strategy could be transformative and impactful in cancer immunotherapy.

## Methods
**Materials.** Ester-terminated PLGA (with inherent viscosity of 0.55-0.75 dL/g) was purchased from Durect Corporation. Lipid PEGs terminated with methoxyl groups (1,2-distearoyl-sn-glycero-3-phosphoethanolamine-*N*-[methoxy(polyethylene glycol) −3000] (ammonium salt), DSPE-MPEG (molecular weight (MW) of PEG, 3000 Da) were purchased from Avanti Polar Lipids. Cationic ethylenediamine core-poly(-amidoamine) (PAMAM) dendrimer generation 0 (G0) were purchased from Sigma-Aldrich. CXCR4-targeting peptide CTCE-9908 (KGVSLSYRCRYSLSVGK, CTCE) and scrambled peptide (LYSVKRSGCGSRKVSYL, SCP) were custom synthesized by GL Biochem (Shanghai) Ltd. Lipofectamine 2000 (L2K) was purchased from Invitrogen. Firefly Luciferase mRNA (Luc mRNA, L-7202), Enhanced Green Fluorescent Protein mRNA (EGFP mRNA, L-7201), and Cyanine 5 Firefly Luciferase mRNA (Cy5-Luc mRNA, L-7702) were purchased from TriLink Biotechnologies (San Diego, CA).

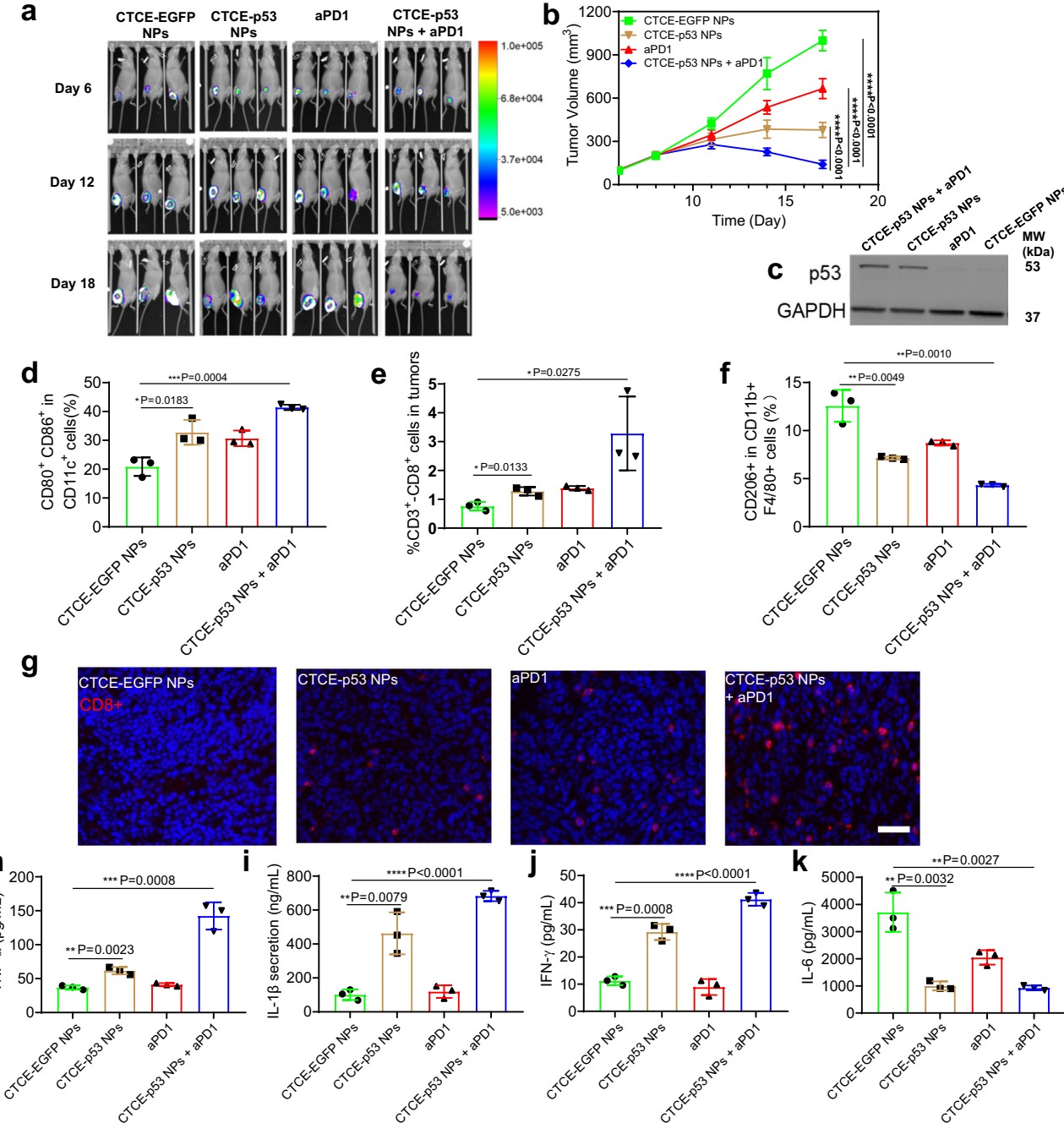

**Fig. 4 Combining CXCR4-targeted p53 mRNA NPs with PD-1 blockade reprograms the immune TME and promotes antitumor immunity in ectopic HCC. a** Bioluminescence images of the luciferase-expressing RIL-175 tumors grafted subcutaneously in C57Bl/6 mice after 6, 12, and 18 days of treatment ($n = 3$ mice/group). **b** Tumor growth rate in each treatment group ($n = 7$ mice/group; ***$P < 0.001$). **c** Western blotting analysis on the expression levels of p53 protein in the s.c. RIL-175 tumors after treatment. GAPDH was used as the loading control. **d**–**f** Flow cytometry analysis ($n = 3$ tumor samples from each group) of lymph node CD80+CD86+ dendritic cells gating on CD11c+ cells (**d**), and tumor-infiltrating CD8+CD3+ T cells (**e**) and M2-like CD206+F4/80+CD11b+ macrophages (**f**). **g** Representative immunofluorescence for CD8 (in red) to confirm intratumoral T cell infiltration after treatment with CTCE-EGFP NPs, anti-PD-1 (aPD1), CTCE-p53 NPs, or the combination. Scale bar: 200 μm. **h**–**k** Protein array analysis of differential expression of cytokines in s.c. HCC tissues after treatment ($n = 3$ samples per group): TNF-α (**h**), IL-1β (**i**), IFN-γ (**j**), and IL-6 (**k**). Statistical significance was calculated using one-way ANOVA with a Tukey post-hoc test. All data are presented as mean ± S.D. For **c** and **g**: this experiment was repeated thrice independently with similar results. *$P < 0.05$; **$P < 0.01$; ***$P < 0.001$; ****$P < 0.0001$. Source data are provided as a Source Data file.

Murine p53 mRNA with chemical modification (full substitution of Pseudo-U and 5-Methyl-C, Capped (Cap 1) using CleanCap® AG, Polyadenylated (120 A)) was custom-synthesized by TriLink Biotechnologies (San Diego, CA). *InVivo*MAb anti-mouse PD-1 (CD279) was purchased from Bioxcell. D-luciferin-K + salt bioluminescent substrate (no. 122799) was obtained from PerkinElmer. Primary antibodies used for western blot experiments as well as immunofluorescent and immunohistochemistry staining included: anti-p53 (sc-126, Santa Cruz Biotechnology, 1:500 dilution), anti-GAPDH (Cell Signaling Technology, # 5174; 1:2000 dilution), anti-beta-Actin (Cell Signaling Technology; 1: 2,000 dilution), and anti-rabbit and anti-mouse horseradish peroxidase (HRP)-conjugated secondary antibodies (Cell Signaling Technology). Secondary antibodies used in this study included: Alexa Fluor® 488 Goat-anti Rabbit IgG (Life Technologies, A-11034), and Alexa Fluor® 647 Goat-anti Mouse IgG (Life Technologies, A-28181). All other chemicals and solvents were purchased from Sigma-Aldrich and used without further purification.

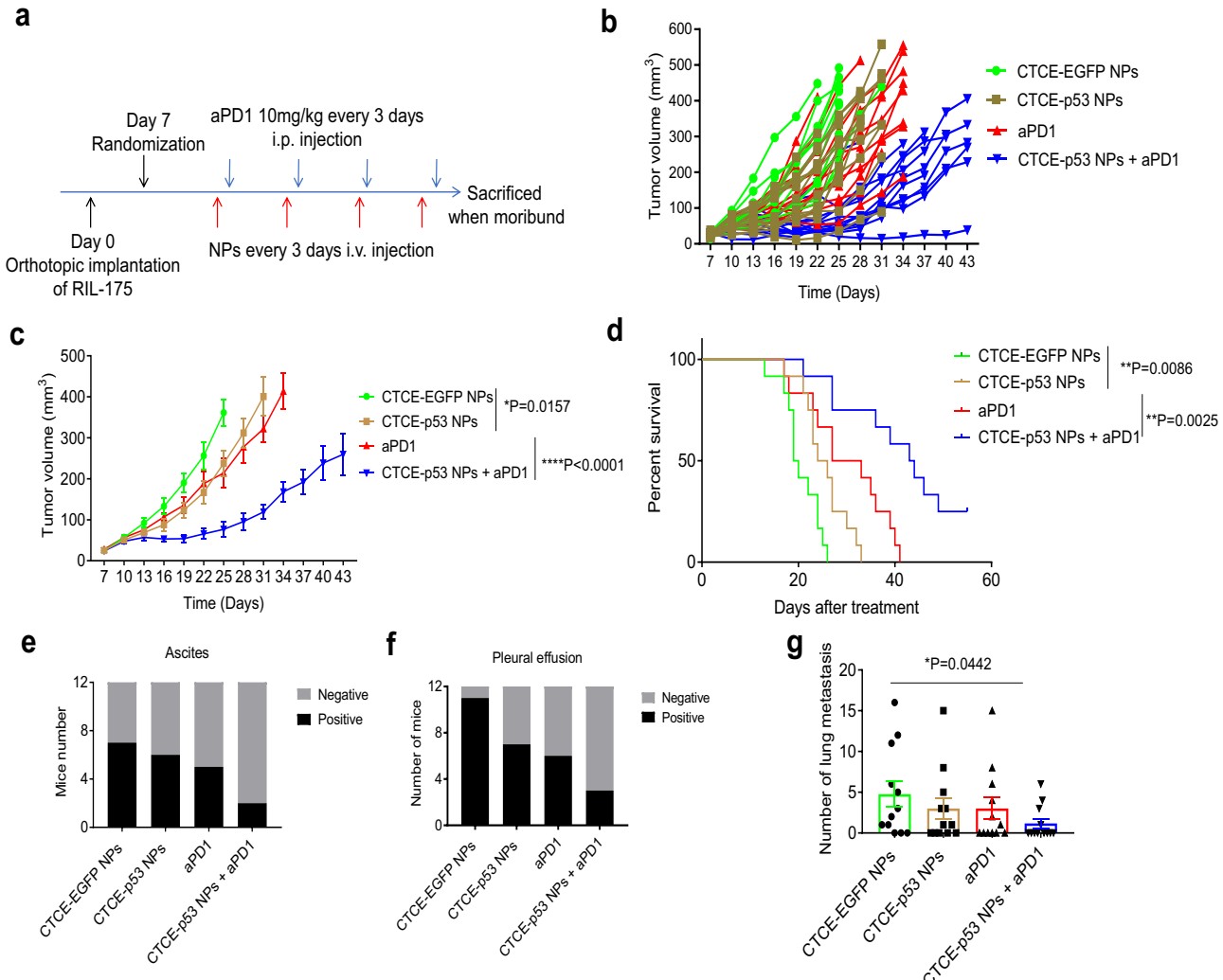

**Fig. 5 Therapeutic efficacy of the combination of CTCE-p53-mRNA NPs with anti-PD-1 (aPD1) in orthotopic HCC model. a** Timeline of tumor implantation and treatment schedule for survival studies in HCC models. **b, c** Tumor growth profile of each indicated treatment group ($n = 12$ mice/group). **d** Survival data from the RIL-175 orthotopic mouse model ($n = 12$ mice/group). **e, f** The combination of CTCE-p53-mRNA NPs with aPD1 reduces ascites (**e**) and pleural effusion (**f**). **g** The combination of CTCE-p53-mRNA NPs with aPD1 reduces lung metastasis ($n = 12$ mice for each group). Statistical significance was calculated via one-way ANOVA with a Tukey post-hoc test. All data are presented as mean ± S.E.M. *$P < 0.05$; **$P < 0.01$. Source data are provided as a Source Data file.

**Synthesis of ionizable lipid-like compounds (G0-Cn).** A series of ionizable lipid-like compounds termed G0-Cn were synthesized through ring opening of epoxides bearing different alkyl chain lengths by generation 0 of poly (amidoamine) (PAMAM) dendrimers (M1). Briefly, substoichiometric amounts of epoxide were added to increase the proportion of products with one less tail than the total possible for a given amine monomer. The amine (1 equiv, typically 1 millimole (mmol)) and epoxide (9 equiv, typically 1 millimole (mmol)) were added to a 50 mL round-bottom glass flask containing a magnetic stir bar. The flask was sealed, and the reaction was heated to 95 °C with homogeneous stirring for 2 days. The crude products were separated by chromatography on silica with gradient elution from $CH_2Cl_2$ to 15:1 $CH_2Cl_2/MeOH$. The separated product was characterized by $^1H$ NMR spectrum.

**mRNA complexation ability of G0-C8 and its stability in organic solvent.** Gel electrophoresis was used to study the mRNA complexation ability of ionizable compound G0-C8 and optimize the ratio between G0-C8 and mRNA in the NPs with free EGFP-mRNA or EGFP-mRNA complexed with G0-C8. Free EGFP-mRNA was also incubated with DMF to evaluate the stability of mRNA in organic solvent (DMF). The EGFP-mRNA were first incubated with G0-C8 at different weight ratios (weight ratios of G0-C8/mRNA: 1, 2, 5, 10, and 20) or DMF for 20 min at room temperature. The volumes of samples were then adjusted with loading dye (Invitrogen) and run into an E-Gel 2% agarose (Invitrogen) gel for 30 min at 50 V. Ambion Millennium markers-Formamide (Thermo Fisher Scientific) was used as a ladder. Finally, the gel was imaged under ultraviolet and the bands were analyzed.

**Synthesis of lipid-PEG-CTCE HCC targeting peptide (DSPE-PEG-CTCE) and lipid-PEG- scrambled peptide (DSPE-PEG-SCP).** We conjugated the CXCR4-targeting peptide CTCE-9908 (KGVSLSYRCRYSLSVGK, CTCE) and scrambled peptide (LYSVKRSGCGSRKVSYL, SCP) to DSPE-PEG-MAL to construct the HCC targeted NPs and the non-targeted control NPs, respectively. Synthesis of DSPE-PEG-CTCE and DSPE-PEG-SCP was achieved through the efficient thiol-maleimide Michael addition click reaction. In brief, DSPE-PEG-maleimide and the thiol-CTCE peptide (3:1) or thiol-scrambled peptide were each dissolved in dimethylsulfoxide (DMF). The peptide solution was diluted in 0.1 M sodium phosphate buffer, pH 7.4, and DSPE-PEG was then added to the mixture. The final reaction mixture was 1:1 DMF/(sodium phosphate buffer) with 5 mM peptide and 15 mM DSPE-PEG maleimide. The reaction was allowed to proceed for 2 h at room temperature and then dialyzed against DI water for purification. Lastly, the product was lyophilized to obtain white powder as the final product (DSPE-PEG-CTCE or DSPE-PEG-SCP). The chemical structures of DSPE-PEG-CTCE and DSPE-PEG-SCP were confirmed by $^1H$-NMR spectrum.

**Optimization of the mRNA NPs: the effect of targeting ligand densities.** The cellular uptake of Enhanced Green Fluorescent Protein mRNA (EGFP mRNA) NPs engineered with seven different densities of CTCE peptide (EGFP-mRNA-CTCE NPs, CTCE density: 2%, 3%, 4%, 5%, 6%, 7%, and 10%, respectively) and 5% scrambled peptide (SCP) was studied to optimize the surface chemistry and targeting efficacy of the mRNA NPs by measuring GFP expression using flow

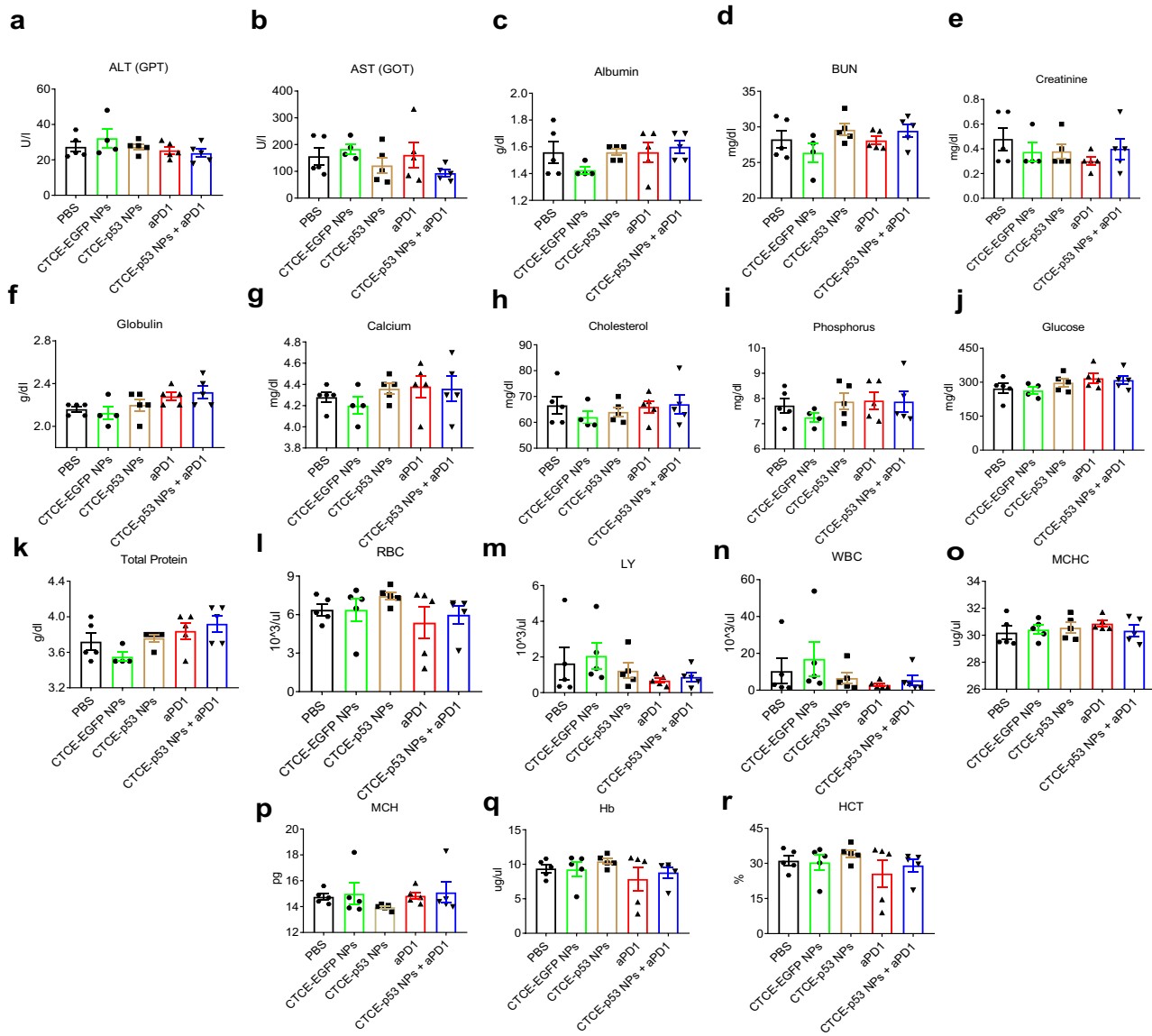

**Fig. 6 In vivo safety of CTCE-p53 NPs and the combination with anti-PD-1 antibody. a–k** Serum biochemistry analysis ($n = 4$ samples for CTCE-EGFP NPs group; $n = 5$ samples for the left four groups). **l–r** Whole blood panel tests analysis ($n = 5$ samples for each group). All data are presented as mean ± S.E.M. Source data are provided as a Source Data file.

cytometry (BD Biosystems, Heidelberg, Germany) and analyzed using Flowjo software (Flowjo V10).

**Preparation of mRNA NPs and the formulation optimization**. An optimized and robust self-assembly technique was employed to prepare mRNA-encapsulated polymer-lipid hybrid NPs based on our previous report[27], but we extensively optimized the ratios among different NPs' components, the pH of the solution for mRNA complexation, and the sequence in which reagents were added, which affected the encapsulation, morphology, and transfection efficiency of the mRNA. Briefly, G0-C8 and PLGA were dissolved separately in anhydrous DMF to form a homogeneous solution at concentrations of 2.5 mg/ml and 5 mg/ml, respectively. DSPE-MPEG, DSPE-PEG-CTCE and DSPE-PEG-SCP were dissolved in DNase/RNase-free HyPure water (GE Healthcare Life Sciences, catalog no. SH30538) at the concentration of 1 mg/mL. All of the reagents listed above were sonicated for 5 min in a water-bath sonicator before use. Citrate buffer with pH 3.0–3.5 was first added to 80 µg of G0-C8 (in 32 µl of DMF), then 16 µg of p53 mRNA (in 16 µl of citrate buffer) was added, mixed gently (at a G0-C8/mRNA weight ratio of 5), and allowed to stay at room temperature for 15 min to ensure the sufficient electrostatic complexation. Afterwards, 250 µg of PLGA polymers (in 50 µl of DMF) was added to the mixture and gently mixed. The final mixture was added dropwise to 10 ml of DNase/RNase-free HyPure water consisting of 1 mg hybrid lipid-PEGs under uniform magnetic stirring (1000 rpm) for 30 min. An ultrafiltration device (EMD Millipore, MWCO 100 kDa) was used to remove the organic solvent and free

compounds from the NP dispersion via centrifugation at 4 °C. After washing 3 times with DNase/RNase-free HyPure water, the mRNA NPs were collected and finally concentrated in pH 7.4 PBS buffer. The NPs were used fresh or stored at −80 °C for further use.

**Physicochemical characterization and stability of mRNA NPs**. The hydro-dynamic diameter, zeta potential, and morphology of the p53-mRNA NPs were measured to assess their physicochemical properties. Sizes and zeta potentials of both CTCE- p53-mRNA NPs and SCP-p53-mRNA NPs were measured by dynamic light scattering (DLS, Brookhaven Instruments Corporation) at 20 °C. Diameters are reported as the intensity mean peak average. To prepare NPs for Transmission Electron Microscopy (TEM) to characterize their morphology and shape, CTCE-p53-mRNA NPs were negatively stained with 2% uranyl acetate and then imaged with a Tecnai G2 Spirit BioTWIN microscope (FEI Company). To verify the in vitro stability of the synthesized polymer-lipid hybrid mRNA NPs in an environment mimicking the physiological milieu, CTCE-p53-mRNA NPs were incubated in 10% serum-containing PBS solution at 37 °C in triplicate for 96 hr with constant stirring at 100 rpm. At each time point, an aliquot of NP solution was withdrawn for particle size measurement using DLS and analyzed at various time intervals to evaluate any change in size distribution. To test the encapsulation efficiency (EE%) of mRNA in the NPs, Cy5-Luc-mRNA NPs were prepared according to the aforementioned method. Dimethyl sulfoxide (DMSO, 100 µl) was added to 5 µl of the NP solution to extract the mRNA encapsulated in the NPs, and

the fluorescence intensity of Cy5-Luc-mRNA was measured using a multi-mode microplate reader (TECAN, Infinite M200 Pro). The amount of loaded mRNA in the engineered NPs was calculated to be ~67.5%.

**Cell culture.** The *p53*-null murine HCC cell line RIL-175 was used throughout. RIL-175 (a *p53*-null/Hras mutant line syngeneic to C57Bl/6 mouse strain background, Luciferase-tagged) was kindly provided by Dr. Tim Greten (NIH). All other cells were purchased from American Type Culture Collection (ATCC). Dulbecco's Modified Eagle's Medium (DMEM; ATCC) was used to culture RIL-175 cells. The cell culture medium was supplemented with 10% fetal bovine serum (Hyclone, SH30071.03), Pen-Strep (100 U ml$^{-1}$ and 100 μg ml$^{-1}$, respectively). Cell culture and all biological experiments were performed at 37 °C in 5% CO$_2$ conditions and the normal level of O$_2$ in a cell culture incubator. All cell lines were routinely tested using a mycoplasma contamination kit (R&D Systems) before any in vitro cell experiments or in vivo tumor model preparation.

**Cell viability and transfection efficiency of EGFP-mRNA NPs.** CTCE-EGFP-mRNA NPs and SCP-EGFP-mRNA NPs were prepared for evaluated the cell viability of the mRNA NPs along with their transfection efficiency of EGFP-mRNA. For the cell viability tests, RIL-175 cells were plated in a 96-well plate at a density of $5 \times 10^3$ cells per well. After 24 h of cell adherence, cells were treated with EGFP-mRNA at various mRNA concentrations (0.0625, 0.125, 0.250, 0.500, and 0.750 μg ml$^{-1}$) for 24 hr, the cells were washed with PBS buffer (pH 7.4), followed by changing the culture medium to 0.1 ml fresh complete medium per well and further incubation for another 24 hr to evaluate cell viability by the Alamar Blue assay according to the manufacturer's protocol and a microplate reader (TECAN, Infinite M200 Pro). To test the transfection efficiency, RIL-175 cells were seeded at a density of $5 \times 10^4$ cells per well on a 6-well plate and allowed to attach and grow until ~80% confluence. Cells were transfected with EGFP-mRNA NPs at the mRNA concentration of 0.5 μg ml$^{-1}$ for 24 h followed by washing with fresh complete medium and further incubated for 24 h to assess transfection efficiency by measuring GFP expression using flow cytometry (DXP11 Flow Cytometry Analyzer). The percentages of GFP-positive cells were calculated and analyzed using Flowjo software (Flowjo V10).

**Establishment of CXCR4-KO RIL-175 cells.** The precise gene-editing system of CRISPR (clustered regularly interspaced short palindromic repeat)/Cas9 (CRISPR associated) was performed to knock out the CXCR4 gene in RIL-175 cells. Briefly, the single guide RNA (sgRNA) targeting CXCR4 was designed on the online tool (http://www.genome-engineering.org) including sgRNA1 (forward: 5′-CACCGTCGAGAGCATCGTGCACAAG-3′, reverse:5′-AAACCTTGTGCAC-GATGCTCTCGAC-3′) and sgRNA 2 (forward: 5′-CACCGGGACTTACACTCA-CACTGAT-3′, reverse: 5′-AAACATCAGTGTGAGTGTAAGTCCC-3′), and sequentially were phosphorylated and annealed. At one time, the lentiviral expression lentiCRISPRv2 plasmid (Addgene, cat. no. 52961, USA) was digested and dephosphorylated with BsmBI enzyme (ThermoFisher, cat. No. ER0451) following by running DNA gel and gel purify the larger band leaving the 2 kb filler piece. Next, the ligation reaction of lentiCRISPRv2 and sgRNAs was established for incubating 10 min at room temperature. After finishing the process of transformation in Stbl3 bacteria and validation by DNA sequencing, the lentiCRISPv2 inserted with sgRNAs targeting CXCR4 was selected out. Then the lentivirus system including lentiCRRISPv2 and the packaging plasmids pVSVg (AddGene, cat. No.8454) and psPAX2 (AddGene, cat. No.12260) were co-transfected into HEK293T cells to produce the complete lentivirus and further transfected into RIL-175 wide type cells. The puromycin (2 μg/μl) previously included in the lenti-CRISPRv2 was used to screen out the positive cells successfully transfected with the complete lentivirus. Finally, the quantitative PCR and western blotting were performed to detect the expression of CXCR4 from both transcriptional and protein levels.

**Cellular uptake of dye-labeled mRNA-encapsulated NPs.** To monitor the cellular uptake of the NPs, Cy5-Luc-mRNA-NPs were prepared. RIL-175 cells were first seeded in 35 mm confocal dishes (MatTek) at a density of $5 \times 10^4$ cells per well and incubated at 37 °C in 5% CO$_2$ for 24 h. The cells were then incubated with medium (DMEM) containing Cy5-Luc-mRNA-NPs at different time intervals. The cells were then washed with PBS, counterstained with Hoechst 33342 (Thermofisher), and analyzed using an Olympus microscope (FV1200, Olympus).

**In vitro cell growth inhibition assay with p53-mRNA NPs.** RIL-175 or HCA-1 cells were plated in 96-well plates at a density of $5 \times 10^3$ cells per well. After 24 h of cell adherence, cells were treated with empty NPs (blank NPs), free p53 mRNA, p53-mRNA NPs at different mRNA concentrations (0.0625, 0.125, 0.250, 0.500, and 0.750 μg ml$^{-1}$). After 24 h of incubation, the cells were washed with PBS buffer (pH 7.4) and further incubated in fresh medium for another 24 h. AlamarBlue cell viability was used to verify the in vitro cell growth inhibition efficacy of p53-mRNA NPs.

**Immunoblotting.** Protein extracts from cells taken from dissected tumors in each group were prepared using lysis buffer (1 mM EDTA, 20 mM Tris-HCl pH 7.6, 140 mM NaCl, 1% aprotinin, 1% NP-40, 1 mM phenylmethylsulphonyl fluoride, and 1 mM sodium vanadate), and supplemented with protease inhibitor cocktail (Cell Signaling Technology) and boiled at 100 °C for 10 min. Equal amounts of protein were determined with a bicinchoninic acid protein assay kit (Pierce/Thermo Scientific) according to the manufacturer's instructions. After gel electrophoresis and protein transformation, membranes were blocked with 3% bovine serum albumin (BSA) in TBST (150 mM NaCl, 50 mM Tris-HCl at pH 7.4, and 0.1% Tween 20) for 1 h at room temperature with gentle shaking. Membranes were rinsed and then incubated overnight at 4 °C with appropriate primary antibodies. The immunoreactive bands were visualized using an enhanced chemiluminescence (ECL) detection system (Cell Signaling Technology).

**Immunofluorescence staining and microscopy.** For immunofluorescence staining, cells or tumor tissues from each treatment group were washed with ice-cold PBS and fixed with 4% paraformaldehyde (Electron Microscopy Sciences) in PBS for 20 min at room temperature, followed by permeabilization in 0.2% Triton X-100-PBS for 10 min. Samples were followed by blocking with PBS blocking buffer containing 2% normal goat serum, 2% BSA, and 0.2% gelatin for 1 h at room temperature. Then, the samples were incubated in primary antibodies at the appropriate concentration for 1 h at room temperature, washed with PBS and incubated in goat anti-rat-Alexa Fluor 647 (Molecular Probes) at 1:1000 dilution in blocking buffer for another 1 h at room temperature. Finally, stained cells were washed with PBS, counterstained with Hoechst 33342 (Molecular Probes-Invitrogen, H1399, 1:10000 dilution in PBS), and mounted on slides with Prolong Gold antifade mounting medium (Life Technologies). The slides were imaged under a confocal laser scanning microscope (Olympus, FV1100).

**Animals.** For the s.c. tumor model, all animal procedures were performed in ethical compliance and with approval by the Institutional Animal Care and Use Committees at Harvard Medical School. Immunocompetent male and female C57BL/6 mice (5-6 weeks old or 6–8 weeks old) were obtained from Charles River Laboratories and housed in a pathogen-free animal facility of Brigham and Women's Hospital, Harvard Medical School. For each experiment, mice were randomly allocated to each group. Mice were put for at least a 72 h acclimation period prior to use in order for physiological parameters to return to baseline after shipping and transferring. All animals were housed in single-unit cages with 12-h alternate light and dark cycles and at controlled ambient temperature (68-79 °F) with humidity between 30%-70%. For the orthotopic tumor model, all animal experiments were performed after approval by the Institutional Animal Care and Use Committee of the Massachusetts General Hospital.

**Pharmacokinetics study.** Healthy C57Bl/6 mice (5–6 weeks old, $n = 3$ per group) were injected intravenously with free Cy5-Luc-mRNA, CTCE-Cy5-Luc-mRNA NPs, or SCP-Cy5-Luc-mRNA NPS through the tail vein at the mRNA dose of 350 μg per kg of animal weight. Blood was collected retroorbitally at different time points (5 min, 30 min, 1 h, 2 h, 6 h, 12 h, and 24 h) and the fluorescence intensity of Cy5-Luc-mRNA was measured using a microplate reader (TECAN, Infinite M200 Pro). Pharmacokinetics was evaluated by calculating the percentage of Cy5-Luc mRNA in blood at various time points.

**HCC tumor model preparation.** Two *p53*-null RIL-175 HCC tumor models, an ectopic (s.c.) grafted model and an orthotopic model, were developed for in vivo biodistribution, modulation of the immune microenvironment, therapeutic efficacy, and in vivo toxicity studies. An orthotopic p53-wild type HCA-1 HCC tumor model was also developed for the in vivo therapeutic efficacy study. For the s.c. grafted model, $~1 \times 10^6$ RIL-175 cells in 100 μl of culture medium mixed with 100 μl of matrigel (BD Biosciences) were implanted subcutaneously in the right flank of C57Bl/6 mice (6–8 weeks old). Mice were monitored for tumor growth every other day according to the animal protocol. To develop the RIL-175 orthotopic model, ~1 million RIL-175 cells 1:1 in Matrigel (Mediatech/Corning, Manassas, VA) were grafted into the left extrahepatic lobe of C57Bl/6 mice (6–8 weeks old). Tumor growth was monitored by high-frequency ultrasonography every 3 days according to the animal protocol. For the HCA-1 orthotopic model, approximately 1 million HCA-1 cells 1:1 in Matrigel (Mediatech/Corning, Manassas, VA) were grafted into the left extrahepatic lobe of C3H mice (6–8 weeks old). Tumor growth was monitored by high-frequency ultrasonography every 3 days according to the animal protocol. When the tumor volume reached about ~100 mm$^3$ (for ectopic model) or ~5 mm in diameter (for orthotopic model), mice were randomly assigned to a treatment group.

**Biodistribution of mRNA NPs in the RIL-175 HCC tumor model.** The biodistribution and tumor accumulation of mRNA NPs were assessed in C57Bl/6 mice bearing with s.c. grafted RIL-175 tumor (~100–200 mm$^3$) and in the RIL-175 orthotopic model (~5 mm in diameter), respectively. In brief, RIL-175 bearing C57Bl/6 mice (5–6 weeks old, $n = 3$ per group) were injected intravenously with free Cy5-Luc-mRNA, CTCE-Cy5-Luc NPs or SCP-Cy5-Luc NPs via the tail vein at a mRNA dose of 350 μg per kg of animal weight. After 24 h, all the mice were

sacrificed, and dissected organs and tumors were visualized using a Syngene PXi imaging system (Synoptics Ltd). The data were analyzed by Image J software.

**Flow cytometry and cytokine analysis**. Tumor immune-environment responses were assessed in the s.c. grafted and orthotopic HCC models by cytokine detection and flow cytometry after treatment. RIL-175 tumor-bearing C57Bl/6 mice (6–8 weeks old, $n = 3$ per group) were systemically (i.v. via tail vein) injected with CTCE-targeted p53 mRNA NPs or control groups (i.e., PBS or CTCE-EGFP NPs) every 3 days for four injections (at the murine p53 or EGFP mRNA dose of 350 μg/kg animal body weight). For the combinatorial immunotherapy group, one day after each i.v. injection of CTCE-p53 NPs, mice underwent intraperitoneal (i.p.) administration of aPD1 (100 μg per dose). The tumor inoculation and treatment schedule are depicted in Fig. 3a and Supplementary Fig. 22a. Forty-eight hrs post treatment, mice were euthanized and tumor tissue was harvested and homogenized for flow cytometry and cytokine analysis. For flow cytometry, tumor tissues were resected and minced, and fragments were incubated in HBSS with 1.5 mg/mL of hyaluronidase and 15 μg/mL of collagenase for 30 minutes at 37 °C. Digested tissues were passed through a 70-μm cell strainer and washed twice with phosphate-buffered saline (PBS)/0.5% bovine serum albumin. Prior to immunostaining, cells were washed with the buffer and fixed and permeabilized with FoxP3/Transcription Factor Staining Buffer Set (eBioscience/Thermo Fischer Scientific) to stain the intracellular markers. Harvested cells were incubated in Dulbecco's Modified Eagle Medium with cell activation cocktail with BD Leukocyte Activation Cocktail, with BD GolgiPlug™(1:500, Biolegend) for 6 h at 37 °C. The cells were stained with the antibodies of cell surface and intracellular marker in the buffer with brefeldin A. Cells were stained with fluorescence-labeled antibodies CD11c (Biolegend, cat. no. 117310, clone N418), CD80 (Biolegend, cat. no. 104722, clone 16-10A1), CD 86 (Biolegend, cat. no. 105005, clone clone GL-1), CD4 (Biolegend, cat. no. 100412, clone GK1.5), CD3 (Biolegend, cat. no. 100204, clone 17 A2), CD8 (Biolegend, cat. no. 140408, clone 53–5.8), CD11b (Biolegend, cat. no. 101208, clone M1/70), F4/80 (Biolegend, cat. no. 123116, clone BM8), CD206 (Biolegend, cat. no. 141716, clone C068C2), Gr-1 (Biolegend, cat. no. 108412, clone RB6-8C5), CD45 (Biolegend, cat. no. 103108, clone 30-F11), TCR (Biolegend, cat. no. 109243, clone H57-597), CD39 (Biolegend, cat. no. 143805, clone Duha59), Ki67 (Biolegend, cat. no. 652423, clone 16A8), CD11b (Biolegend, cat. no. 101243, clone M1/70), CD206 (Biolegend, cat. no. 141717, clone C068C2), Forkhead box protein P3 (FoxP3; Biolegend, cat. no. 126419, clone MF-14), IFN-γ Receptor βchain (Biolegend, cat. no. 113605, clone MOB-47), CD119 (BD Bioscience, cat. no. 740897, clone GR20), FITC (Biolegend, cat. no. 503805, clone JES6-5H4) following the manufacturer's instructions. All antibodies were diluted 200 times, except for FoxP3 and CD119 staining, which were 1:100 dilution. The stained cells were measured on a flow cytometer (Accuri C6 Plus, BD Biosciences) and analyzed by FlowJo software (Flowjo V10). The numbers presented in the flow cytometry analysis images are percentage based. For cytokine studies, tissue samples were assayed in duplicate using the MSD proinflammatory Panel I, a highly sensitive multiplex enzyme-linked immunosorbent assay (ELISA) for quantitatively measuring 10 cytokines-IFN-γ, interleukin (IL)−1β, IL-2, IL-4, IL-5, IL-6, IL-10, IL-12p70, TNF-α, KC/GRO and IL-9, IL-15, IP-10, MCP-1, MIP-1α, MIP-2, IL-17A/F, IL-27p28/IL-30, IL-33 using electrochemiluminescence-based detection (MesoScale Discovery, Gaithersburg, MD).

**In vivo therapeutic efficacy**. The therapeutic effects of p53-mRNA NPs and their integrated antitumor effect with anti-PD1 were evaluated in the *p53*-null HCC s.c. RIL-175 tumor model, *p53*-null RIL-175 orthotopic tumor model, and *p53*-wild-type HCA-1 orthotopic tumor model. For the s.c. model, RIL-175 tumor-bearing C57Bl/6 mice (6–8 weeks old, $n = 5$ per group) were monitored for tumor growth every other day after tumor implantation; tumor size was measured using a digital caliper and calculated as $0.5 \times \text{length} \times \text{width}^2$. When the tumor volume reached about ~100 mm³, mice were randomly divided into five groups ($n = 5$), which received treatment with PBS, CTCE-EGFP NPs, CTCE-p53 NPs, aPD1, or the combination of CTCE-p53 NPs and aPD1 according to the schedule in Supplementary Fig. 22a at the mRNA dose of 350 μg/kg animal body weight, while the aPD1 were administrated by i.p. at 100 μg per dose one day after the p53-mRNA NPs treatment. Tumor growth was measured and calculated every 3 days. The body weights of all mice were recorded every three days during this period. Animals were euthanized upon showing signs of imperfect health or when the size of their accumulated tumors exceeded 1.0 cm³. For the orthotopic HCC tumor model, tumor growth was monitored by high-frequency ultrasonography every 3 days. When the tumor size reached ~5 mm in diameter, mice were randomly assigned to a treatment group ($n = 12$). Treatments were administered according to the schedule in Fig. 3a. For the comparison of side-by-side the in vivo survival of the combination of CTCE-p53 NPs with aPD1 against the new standard of care in HCC patients (i.e., anti-VEGFR2 antibody + aPD-L1 antibody) in the orthotopic RIL-175 tumor model, treatments were administered i.p. every 3 days for 4 doses at 10 mg/kg of aPD-L1 antibody (Bioxcell, #BE0101, clone 10F.9G2), and 10 mg/kg of anti-VEGFR-2 antibody (Bioxcell, #BE0060, clone DC101) (Supplementary Fig. 20a). For survival studies, the endpoint was moribund status, defined as signs of prolonged distress, >15% weight loss compared with the starting date, body condition score >2, or tumor size of >15 mm in diameter.

**Bioluminescence**. To further explore the therapeutic efficacy of our therapeutic strategy, tumors were also assessed using an in vivo bioluminescence imaging system (Bruker Xtreme scanner). Mice were monitored for tumor growth by bioluminescent in vivo imaging every 6 days (Day 0, 6, and 12); specifically, 8 minutes after intraperitoneal injection of 150 mg/kg D-luciferin substrate (PerkinElmer, Catalog#122799), mice from each treatment group ($n = 3$) were imaged.

**Immunohistochemistry staining**. The expression of p53 protein and CD8 + cells in tumor tissue sections from different in vivo treatment groups were assessed by immunohistochemistry. Tumor sections were fixed in 4% buffered formaldehyde solution and embedded in paraffin. Paraffin-embedded sections were deparaffinized, rehydrated, and washed in distilled water. In order to retrieve the antigen, tumor tissue sections were incubated in 10 mM citrate buffer (pH = 6) for 30 min, washed in PBS, and immersed in 0.3% hydrogen peroxide ($H_2O_2$) for 20 min, then incubated in blocking buffer (5% normal goat serum and 1% BSA) for 60 min. Tissue sections were then incubated with the appropriate primary antibodies (PBS solution supplemented with 0.3% Triton X-100) at 4 °C overnight in a humid chamber. After being rinsed with PBS, the samples were incubated with biotinylated secondary antibody at room temperature for 30 min, rinsed again with PBS, and incubated with the avidin-biotin-horseradish peroxidase complex (ABC kit, Vector Laboratories, Inc). After being washed again, stains were processed with the diaminobenzidine peroxidase substrate kit (Impact DAB, Vector Laboratories, Inc) for 3 min. Sections were evaluated using a Leica Microsystem after being counterstained with hematoxylin (Sigma), dehydrated, and mounted.

**In vivo toxicity evaluation**. The in vivo toxicity of p53-mRNA NPs was comprehensively studied in both the *p53*-null HCC s.c. graft tumor model and the *p53*-null orthotopic HCC tumor model. In brief, the major organs were harvested at the end point, sectioned, and H&E stained to evaluate the histological differences. In addition, blood was drawn, and serum was isolated at the end of the in vivo efficacy experiment. Various parameters including ALT, AST, BUN, RBC, WBC, Hb, MCHC, MCH, HCT, and LY were tested to evaluate toxicity.

**Statistical analysis**. A two-tailed Student's t-test or a one-way analysis of variance (ANOVA) was performed when comparing two groups or more than two groups, respectively. Statistical analysis was carried out using Prism 8.0 (GraphPad) and Microsoft Excel. Data are expressed as standard deviation (S.D.) or standard error means (S.E.M) as described in the main text. Difference was considered to be significant if $P < 0.05$ (*$P < 0.05$, **$P < 0.01$, ***$P < 0.001$, ****$P < 0.0001$ unless otherwise indicated). All studies were performed at least in triplicate unless otherwise stated.

**Reporting summary**. Further information on research design is available in the Nature Research Reporting Summary linked to this article.

## Data availability
The authors declare that all data supporting the findings of this study are available within the Article, Supplementary Information or Source Data file. Source data are provided with this paper.

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

## Acknowledgements

This work was supported by the U.S. Department of Defense (DoD) Peer Reviewed Cancer Research Program (PRCRP) Idea Award with Special Focus (W81XWH1910482) (to J.S and D.G.D.). D.G.D.'s research is also supported by NIH grants R01CA260872 and R01CA260857, and by Department of Defense grants #W81XWH-19-1-0284 and W81XWH-21-1-0738. The authors thank Dr. Peigen Huang, Sylvie Roberge, and Anna Khachatryan (MGH) for outstanding experimental support.

## Author contributions

J.S. and D.G.D. conceived the idea and directed the project. Y.X., J.C., H.Z., X.Z., Z.R., Z.P., X.J., A.M., L.Z., Z.A., D.S.C., and X.H. performed the experiments and analyzed data. Y.X., J.C., D.G.D., and J.S. wrote the manuscript. All authors discussed the results and assisted in the preparation of the manuscript.

## Competing interests

D.G.D. received consultant fees from Bayer, BMS, Simcere, Sophia Biosciences, Innocoll and Surface Oncology and has received research grants from Bayer, Merrimack, Exelixis, Surface Oncology and BMS. No reagents or support from these companies was used for this study. No potential conflicts of interest were disclosed by other authors.
