## [Peer Review File · Nature Communications]

Reviewers' Comments:

Reviewer #1:

Remarks to the Author:

Xiao et al. developed p53 mRNA nanotherapy for hepatocellular carcinoma. They show that expression of p53 enhances anti-tumor immune microenvironment and increases efficiency of treatment when combined with anti-PD1 antibody. The design of the nanoparticles is interesting as the design allow improved targeting of the HCC cells that are known to express CXCR4. Using mouse models (orthotopic and skin xenografts), they show significant anti-tumor activities of the combination therapy that includes p53 mRNA nanoparticle + anti-PD1 antibody. Authors also performed toxicity and pharmacokinetics studies of the p53 mRNA nanoparticles.

After therapy, however, HCC reappears. Nevertheless, they see a significant increase in survival post-therapy. Whereas the ideas are interesting, a major concern is that the ms. did not address how their treatments compare with the standard treatments of HCC.

Major comments:

1. Studies by Shigeta et al. (J. Immunother of Cancer, 2020) using a combination therapy of Regorafenib and anti-PD1 antibody observed similar survival benefits. Regorafenib is used following Sorafenib to treat HCC. It will be important to compare side-by-side whether authors combination is more beneficial than combining anti-PD1 with Sorafenib or Regorafenib.
2. Mouse models were generated with RIL-175 cell line. Unfortunately, all studies were performed on one cell line.
3. P53+ HCC express p53 at low levels. It would important to see whether the combination works on p53+ HCC.

Reviewer #2:

Remarks to the Author:

This is a high-quality work with great innovation. There are a few things that can be done to enhance the feasibility and reliability:

- 1) a CXCR4 KO cell model can be established to verify the selectivity of the mRNA-peptide nanoparticle.
- 2) anti-PD-1 effect is closely correlated with tumor PD-L1 expression. The PD-L1 level on tumors of all treatment groups should be checked.
- 3) There can be more sophisticated molecular studies characterizing how p53 overexpression changes tumor immunogenicity in their model. Some studies have discussed how p53 modulates the immune response. For example, p53 may enhance antigen presentation (DOI: 10.1038/ncomms3359), activates innate immune signaling (DOI: 10.1016/j.ccell.2021.01.003), etc.

Reviewer #3:

Remarks to the Author:

In this manuscript, the authors optimized and developed a lipid-polymer hybrid platform for p53 mRNA delivery to induce p53 expression in HCC. They found the combination of p53 overexpression and anti-PD-1 therapy effectively reprograms the tumor immune microenvironment and improved the anti-tumor effects compared to anti-PD-1 therapy or therapeutic p53 expression alone. Overall, this is a promising strategy to enhance the anti-tumor activity of immune therapy. The major conclusions are fully demonstrated. I would suggest its publication on Nature communication after addressing the following minor comments.

1. The schematic illustration in Figure 1a should clearly present how p53 overexpression enhanced the effect of reprogramming the tumor immune microenvironment.
2. All the confocal imaging, H&E staining and Immunohistochemistry (IHC) staining results should

give out scale bars. E.g. Figure 1c, 2b&g, 3e, 4g, S10, S13c, S18 and S19.

3. The authors should characterize all the Nano formulations (luc, GFP, P53 NPs) used in their study. This reviewer wants to know why the authors selected the Cy5-Luciferase mRNA not Cy5-P53 mRNA for bio-distribution study. Is there any difference?

4. Please check the order of the legend in Figure S13b and Figure S13c.

5. Whether the overexpression of P53 could induce the changes of downstream signaling pathway? This is an important issue.

6. In figure 2c, the authors used CTCE-Cy5 EGFP NPs for circulation profile investigation, but the label is CTCE-Cy5-luc NPs. Please check it.

7. Some detail issues should be carefully checked to improve the quality of figures. E.g. mm3 not mm3.

Reviewer #4:

Remarks to the Author:

In this manuscript, Shi and co-authors developed a CTCE-modified nanoparticle containing p53 mRNA (CTCE-p53 NP). CTCE-p53 NPs enhanced targeted delivery of p53 mRNA into hepatocellular carcinoma cells *in vivo*, which reprogrammed the tumor microenvironment, including activation of CD8+ T cells, polarization of tumor-associated macrophages towards the anti-tumor phenotype, and production of anti-tumor cytokines. These effects led to improved anti-tumor effects, such as survival and tumor volumes, compared to anti-PD-1 therapy or therapeutic p53 expression alone in mouse models. Moreover, no obvious safety concerns were observed at the end of this study by pathological analysis of the mouse blood and major organs. Overall, these experiments support the hypothesis and provide a novel strategy to reverse the immunosuppressive tumor microenvironment, which shows great promise for future clinical applications.

1. As shown in Figure 1d and Figure S4, G0-C8 NP has better mRNA delivery efficacy than other NPs. What might be possible mechanisms behind this (mRNA encapsulation efficiency, cellular uptake, or endosome escape ability)? It would be helpful to include some descriptions.

2. It would be more informative to study mRNA delivery in the study of the stability of NPs in Figure S5b.

3. In Figure 3 and Figure 4, the markers used to identify M1 and M2 macrophages should be added into the legends.

POINT BY POINT RESPONSE TO REVIEWERS' COMMENTS

Reviewer #1 (Remarks to the Author in italics)

Xiao et al. developed p53 mRNA nanotherapy for hepatocellular carcinoma. They show that expression of p53 enhances anti-tumor immune microenvironment and increases efficiency of treatment when combined with anti-PD1 antibody. The design of the nanoparticles is interesting as the design allow improved targeting of the HCC cells that are known to express CXCR4. Using mouse models (orthotopic and skin xenografts), they show significant anti-tumor activities of the combination therapy that includes p53 mRNA nanoparticle + anti-PD1 antibody. Authors also performed toxicity and pharmacokinetics studies of the p53 mRNA nanoparticles.

After therapy, however, HCC reappears. Nevertheless, they see a significant increase in survival post-therapy. Whereas the ideas are interesting, a major concern is that the ms. did not address how their treatments compare with the standard treatments of HCC.

We thank the reviewer for finding our work interesting. We have addressed his/her concern by adding additional data comparing the intervention with standard therapy, as described below.

Major comments:

1. Studies by Shigeta et al. (J. Immunother of Cancer, 2020) using a combination therapy of Regorafenib and anti-PD1 antibody observed similar survival benefits. Regorafenib is used following Sorafenib to treat HCC. It will be important to compare side-by-side whether authors combination is more beneficial than combining anti-PD1 with Sorafenib or Regorafenib.

We agree with the reviewer's concern regarding a comparison with other relevant combinations. However, we were unable to compare side by side the p53 mRNA nanoparticle + anti-PD-1 antibody (aPD1) versus aPD1 with Sorafenib or Regorafenib because that work was performed under a sponsored research agreement with Bayer which precluded us from performing such comparisons. Moreover, the anti-PD1 with Sorafenib or Regorafenib combinations remain experimental at this time, as they have not been clinically validated yet. Instead, we performed a survival study comparing the p53 mRNA NPs + aPD1 versus anti-VEGFR2 antibody + aPD-L1 antibody and versus IgG control in the orthotopic RIL-175 model. Dual antibody blockade of VEGF/PD-L1 pathway is the current clinical standard of care for advanced human HCC. As shown in new Supplementary Figure S20 (included below), p53 mRNA NPs + aPD1 and anti-VEGFR2 Ab + aPD-L1 Ab combinations were both effective and comparable in increasing overall survival and delaying disease morbidity in the p53-KO murine HCC model.

Figure S20. The comparison of the therapeutic efficacy between the combination of CTCE-p53-mRNA NPs with anti-PD-1 (aPD1) and the combined treatment of anti-PD-L1 (aPD-L1) and anti-VEGFR2 (DC101) in orthotopic HCC model in C57BL/6 mice. **(a)** Timeline of tumor implantation and treatment schedule for survival studies in RIL-175 orthotopic murine HCC model. To develop the RIL-175 orthotopic model, approximately 9×10^5 RIL-175 cells 1:1 in Matrigel (Mediatech/Corning, Manassas, VA) were grafted into the left extrahepatic lobe of C57Bl/6 mice (6-8 weeks old). **(b, c)** Tumor growth kinetics in each treatment group measured by ultrasound imaging. **(d)** Survival distributions in each treatment group. For Figures b-d, day 0: the time for the first treatment. VEGFR2: Vascular Endothelial Growth Factor Receptor 2. Statistical significance was analyzed via one-way ANOVA with a Tukey post-hoc test. The statistical method for survival analysis is Logrank test. * $P < 0.05$; ** $P < 0.01$; **** $P < 0.0001$; $n = 12$ mice.

2. Mouse models were generated with RIL-175 cell line. Unfortunately, all studies were performed on one cell line.

We have previously demonstrated the effects of p53 restoration in multiple human cancer cell lines (*Science Translational Medicine* 11, eaaw1565 (2019)). The immunotherapy approach required a syngeneic model, and thus we used RIL-175, which was the only p53-KO model available, to our knowledge. Nevertheless, in response to this comment, we studied the *in vitro* cell viability of p53-wild type murine HCC cell line HCA-1 and performed an *in vivo* survival study using an HCA-1 orthotopic model in C3H mice. As shown in the new Supplementary Figure S18 (copied below), our CTCE-p53 NPs showed a mild (~35%) reduction of cell viability against HCA-1 cells at the high mRNA concentration of 0.5 $\mu\text{g/mL}$, while the control NPs and empty NPs showed negligible cytotoxicity. Unfortunately, this modest *in vitro* effect did not translate into a statistically significant survival benefit (data shown in Supplementary Figure S21, copied below) with the same dosage and dosing frequency used in the RIL-175 model.

Figure S18. HCA-1 cell viability after treatment with control (saline), Empty NPs, Control NPs (CTCE-EGFP NPs), or CTCE-p53 NPs with different mRNA concentrations (0.25 and 0.5 $\mu\text{g/mL}$, respectively).

Figure S21. The therapeutic efficacy of the combination of CTCE-p53-mRNA NPs with anti-PD-1 (aPD1) in orthotopic HCA-1 HCC model in C3H mice. **(a)** Timeline of tumor implantation and treatment schedule for survival studies in murine HCA-1 HCC model. **(b)** Survival distributions in each treatment group. **(c, d)** Tumor growth kinetics in each treatment group. Statistical significance was analyzed via one-way ANOVA with a Tukey post-hoc test. * $P < 0.05$; ** $P < 0.01$; **** $P < 0.0001$; $n = 12$ mice.

3. P53+ HCC express p53 at low levels. It would important to see whether the combination works on p53+ HCC.

We agree with the reviewer. As described above, we performed these experiments. While data showed no anti-tumor effect of CTCE-p53 NPs alone or its combination with aPD1 *in vivo* in the p53-WT model, more in depths studies will be needed to definitively evaluate the efficacy and safety of this combination with higher doses of p53 mRNA NPs across p53-WT tumors, which is beyond the scope of our current study.

Reviewer #2

This is a high-quality work with great innovation. There are a few things that can be done to enhance the feasibility and reliability:

We thank the reviewer for the evaluation. We have addressed the concerns by adding additional data, as described below.

1) a CXCR4 KO cell model can be established to verify the selectivity of the mRNA nanoparticle.

We agree with the reviewer. To further evaluate the selectivity of the CTCE-mRNA NPs, we first examined the targeting effect of CTCE peptide by blocking the CXCR4 receptor on cell surface using free CTCE peptide. As shown in the below Supplementary Fig. S13, after blocking the CXCR4 receptor on RIL-175 cells, the fluorescence intensity of RIL-175 cells co-incubated with CTCE-Cy5-Luciferase mRNA NPs was significantly lower than that without blocking. The finding suggests that the binding of CTCE-Cy5-Luciferase mRNA NPs to RIL-175 cells was effectively blocked by free CTCE peptide.

Fig. S13. (A) Fluorescent images of RIL-175 cells treated with CTCE-Cy5-Luciferase mRNA NPs. (B) Fluorescent images of RIL-175 cells after blocking the CXCR-4 receptor.

Furthermore, we generated a CXCR4-KO RIL-175 cell line (using CRISPR/Cas9 editing) and performed *in vitro* cellular uptake studies. As evidenced by Western blotting in Supplementary Fig. S14 (copied below), CXCR4 expression of the RIL-175 cells were effectively knocked out by CRISPR/Cas9 editing. *In vitro* cellular uptake study (Fig. S15) showed that the fluorescence intensity of CXCR4-KO RIL-175 cells (sgRNA2) co-incubated with Cy5-Luciferase mRNA NPs was significantly reduced than that of the sgControl RIL-175 cells (without CXCR4-knockout), further demonstrating the active targeting effect of the CTCE-NPs.

Fig. S14. Western blotting of the CXCR4 expression of the CXCR4-KO RIL-175 cells (sgRNA1 and sgRNA2) by CRISPR/Cas9 editing.

Fig. S15. Fluorescent images of (A) sgControl RIL-175 cells and (B) CXCR4-KO RIL-175 cells treated with CTCE-Cy5-Luciferase mRNA NPs.

2) anti-PD-1 effect is closely correlated with tumor PD-L1 expression. The PD-L1 level on tumors of all treatment groups should be checked.

As suggested, we now tested PD-L1 expression in RIL-175 cells *in vitro* and RIL-175 tumors *in vivo* after p53 mRNA NPs treatment. Results shown in the Figures R1-R3 demonstrated the elevated level of PD-L1 expression induced by p53 treatment compared to the control groups (EGFP mRNA NPs) both *in vitro* and *in vivo*. These interesting results warrant future studies of understanding how p53 restoration induces PD-L1 expression in tumor cells and combining p53 mRNA NPs with anti-PD-L1 therapy for HCC treatment.

Fig. R1. *In vitro* PD-L1 expression in RIL-175 cells after treatment with PBS (C), CTCE-p53 NPs (p53 NP) at p53 mRNA concentrations of 0.25 µg/mL, 0.5 µg/mL, or 1 µg/mL, and CTCE-EGFP NPs (EGFP NPs) at EGFP mRNA concentrations of 1 µg/mL.

Fig. R2. *In vivo* PD-L1 expression in s.c. RIL-175 tumors after CTCE-p53 NPs or CTCE-EGFP NPs treatment.

PD-L1 DAPI. Bar value: 500 μm

Fig. R3. Representative immunofluorescence (IF) staining of tumor tissues from orthotopic HCC model in C57BL/6 mice using RIL-175 cells after treatment with PBS (control), CTCE-EGFP NPs, CTCE-p53 NPs, aPD1, or CTCE-p53 NPs + aPD1. * $P < 0.05$; *** $P < 0.001$; **** $P < 0.0001$. The below histogram represented the positive PD-L1 cells in various treatment groups.

3) There can be more sophisticated molecular studies characterizing how p53 overexpression changes tumor immunogenicity in their model. Some studies have discussed how p53 modulates the immune response. For example, p53 may enhance antigen presentation (DOI: 10.1038/ncomms3359), activates innate immune signaling (DOI: 10.1016/j.ccell.2021.01.003), etc.

We agree with the reviewer and as suggested, studied the role of p53 on MHC class I expression by Western blotting and immunofluorescence. The results shown in the new Supplementary Fig. S24 and S25 (copied below) revealed an association between p53 and MHC class I expression. Specifically, MHC-I level was increased after p53 mRNA NPs treatment.

Fig. S24. MHC-1 expression in RIL-175 tumor cell line after CTCE-p53 NPs treatment.

Fig. S25. Immunofluorescence images of MHC class 1 expression in RIL-175 cells after p53 mRNA NPs treatment at mRNA concentration of 0.25 µg/mL and 0.5 µg/mL, respectively.

Reviewer #3

In this manuscript, the authors optimized and developed a lipid-polymer hybrid platform for p53 mRNA delivery to induce p53 expression in HCC. They found the combination of p53 overexpression and anti-PD-1 therapy effectively reprograms the tumor immune microenvironment and improved the anti-tumor effects compared to anti-PD-1 therapy or therapeutic p53 expression alone. Overall, this is a promising strategy to enhance the anti-tumor activity of immune therapy. The major conclusions are fully demonstrated. I would suggest its publication on Nature communication after addressing the following minor comments.

We appreciate the evaluation and the positive feedback. The point-by-point responses are given below to address the important points raised by the reviewer.

1. The schematic illustration in Figure 1a should clearly present how p53 overexpression enhanced the effect of reprogramming the tumor immune microenvironment.

We thank the reviewer for this suggestion. We have added the discussion of the underlying mechanism for p53 enhanced the effect of reprogramming the tumor immune microenvironment in the figure caption of Figure 1a as follows: "The combination of CTCE-p53 NPs and PD-1 blockade effectively and globally reprogrammed the immune TME of HCC, as indicated by activation of CD8+ T cells and NK cells, favorable polarization of TAMs towards the anti-tumor phenotype, and increased expression of MHC-I and anti-tumor cytokines."

It has been previously reported that p53 could also play an important role in the suppression of pro-tumorigenic M2-type tumor-associated macrophage (TAM) polarization, thus facilitating antitumor immunity (*Int J Cancer* **145**, 2535-2546 (2019); *Cell death and differentiation* **22**, 1081-1093 (2015)). Our data show that p53 could also upregulate the expression of the major histocompatibility complex (MHC) class I molecules expression on tumor cells, which further support the immune activation by this approach (data shown in the new Supplementary Figs. S24 and S25).

2. All the confocal imaging, H&E staining and Immunohistochemistry (IHC) staining results should give out scale bars. E.g. Figure 1c, 2b&g, 3e, 4g, S10, S13c, S18 and S19.

Per suggestion, we have added scale bars to all the figures in our revised manuscript.

3. The authors should characterize all the Nano formulations (Luc, GFP, P53 NPs) used in their study. This reviewer wants to know why the authors selected the Cy5-Luciferase mRNA not Cy5-P53 mRNA for bio-distribution study. Is there any difference?

We thank the reviewer for this important comment. We have characterized all the NP formulations used in this study, including Luc mRNA NPs, GFP mRNA NPs, and p53 mRNA NPs. As shown in the Supplementary Fig. S6 below, all the NP formulations exhibited similar average size and zeta potential. Since Cy5-p53 mRNA is not commercially available and luciferase mRNA NPs and p53 mRNA NPs have similar particle size and zeta potential, we therefore used Cy5-Luciferase mRNA for the biodistribution study. We expect that the biodistribution result using Cy5-Luciferase mRNA may be representative, given the similarity of particle size, surface charge, and formulation of these NPs. It is worth noting that both dye-labeled Luc mRNA and dye-labeled GFP mRNA have been frequently used for localization, PK study, *in vivo* biodistribution study, and *in vivo* imaging by us and others, e.g., *Nature Biomedical Engineering* **2**, 850-864 (2018); *Science Translational Medicine* **11**, eaaw1565 (2019); *Nature Communications* **10**, 4333 (2019); *Advanced Functional Materials*, 2011068 (2021).

Fig. S6. (A) Average particle size (nm) and (B) zeta potential (mV) of the CTCE-Luc NPs, CTCE-GFP NPs, CTCE-p53 NPs, CTCE-Cy5-Luc NPs and CTCE-Cy5-GFP NPs (n=3).

4. Please check the order of the legend in Figure S13b and Figure S13c.

We thank the reviewer for pointing out this error, which is now corrected in the revised Supplementary Figure S22.

5. Whether the overexpression of P53 could induce the changes of downstream signaling pathway? This is an important issue.

In the revised manuscript, we provide data on the effect of p53 restoration on MHC class I expression by Western blotting and immunofluorescence. The results shown in the Supplementary **Figs. S24** and **S25** (copied below) demonstrated the association between p53 and MHC class I expression. In addition, our previous work has shown that p53 restoration could inhibit the activation of autophagy and activate apoptosis, see *Science Translational Medicine* 11(523): eaaw1565 (2019).

Fig. S24. MHC-1 expression in RIL-175 tumor cell line after CTCE-p53 NPs treatment.

Fig. S25. Immunofluorescence images of MHC class 1 expression in RIL-175 cells after p53 mRNA NPs treatment at mRNA concentration of 0.25 µg/mL and 0.5 µg/mL, respectively.

6. In figure 2c, the authors used CTCE-Cy5 EGFP NPs for circulation profile investigation, but the label is CTCE-Cy5-luc NPs. Please check it.

We used CTCE-Cy5-Luc NPs for circulation profile investigation, and we have corrected the label to CTCE-Cy5-Luc NPs in the revised Figure 2c accordingly. We thank the reviewer for pointing our error.

7. Some detail issues should be carefully checked to improve the quality of figures. E.g. mm3 not mm3.

We appreciate this comment and have checked all the figures and made changes accordingly.

Reviewer #4

In this manuscript, Shi and co-authors developed a CTCE-modified nanoparticle containing p53 mRNA (CTCE-p53 NP). CTCE-p53 NPs enhanced targeted delivery of p53 mRNA into hepatocellular carcinoma cells in vivo, which reprogrammed the tumor microenvironment, including activation of CD8+ T cells, polarization of tumor-associated macrophages towards the anti-tumor phenotype, and production of anti-tumor cytokines. These effects led to improved anti-tumor effects, such as survival and tumor volumes, compared to anti-PD-1 therapy or therapeutic p53 expression alone in mouse models. Moreover, no obvious safety concerns were

observed at the end of this study by pathological analysis of the mouse blood and major organs. Overall, these experiments support the hypothesis and provide a novel strategy to reverse the immunosuppressive tumor microenvironment, which shows great promise for future clinical applications.

We thank the reviewer for the positive evaluation.

1. As shown in Figure 1d and Figure S4, G₀-C₈ NP has better mRNA delivery efficacy than other NPs. What might be possible mechanisms behind this (mRNA encapsulation efficiency, cellular uptake, or endosome escape ability)? It would be helpful to include some descriptions.

We appreciate this important comment. To study the possible mechanism(s) underlying the effect of G₀-C_n on mRNA delivery, we studied the mRNA encapsulation efficiency and cellular uptake of the mRNA NPs formulated with different cationic lipid-like materials (G₀-C_n). As shown in the Supplementary Table S1 below, different cationic lipid-like materials G₀-C_n showed negligible effect on the encapsulation efficacy of mRNA NPs. However, their effect on cellular uptake seemed to play an important role for the mRNA delivery efficacy, as shown in the Supplementary Fig. S5 below. G₀-C₈ NP showed the highest cellular uptake among all the other G₀-C_n NPs. We have included the new results and discussion in the revised manuscript.

Table S1. Effect of different cationic lipid-like materials G₀-C_n on the encapsulation efficacy of Cy5-Luciferase mRNA NPs.

	G ₀ -C ₈	G ₀ -C ₁₀	G ₀ -C ₁₂	G ₀ -C ₁₄	G ₀ -C ₁₆
Encapsulation efficiency(EE) %	67.3	65.9	63.7	66.9	58.3

Fig. S5. Effect of different cationic lipid-like materials G₀-C_n on the cellular uptake of Luc-mRNA NPs (mRNA concentration: 0.25 µg/mL).

2. It would be more informative to study mRNA delivery in the study of the stability of NPs in Figure S5b.

We agree that it will be more informative to study mRNA delivery to assess the stability of NPs. Thus, we studied the cell viability against RIL-175 cells after treatment with p53-mRNA NPs incubated with 10% serum at various time points up to 96 hr (at 37°C). As shown in the below Supplementary Fig. S8., the p53-mRNA NPs showed comparable cell viabilities in all the groups, which further indicated the stability of the p53-mRNA NPs.

Fig. S8. Stability of p53-mRNA NP in 10% serum at 37°C evaluated by measuring cell viabilities towards RIL-175 cells. The p53-mRNA NPs were first incubated with 10% serum for various time points up to 96 h and then used for treatment.

3. In Figure 3 and Figure 4, the markers used to identify M1 and M2 macrophages should be added into the legends.

We have added the markers for identifying M1 and M2 macrophages into the legends in the revised Figures 3 and 4 (for M1, we used MHC-II; for M2, we used CD206).

Reviewers' Comments:

Reviewer #1:

Remarks to the Author:

No more comments.

Reviewer #2:

Remarks to the Author:

There is a significant improvement in the current revised manuscript. And the authors addressed the questions that I had.

Reviewer #3:

Remarks to the Author:

The authors have done significant improvement and addressed most, if not all, issues. This reviewer has no more questions. Congratulations for generating a great piece of work.

Reviewer #4:

Remarks to the Author:

In this revised manuscript, the authors provided additional results and descriptions to examine the nanoparticles. New experimental data support the project design and conclusions.

Response to REVIEWERS' COMMENTS

Reviewer #1 (Remarks to the Author):

No more comments.

We appreciate the evaluation and the positive feedback.

Reviewer #2 (Remarks to the Author):

There is a significant improvement in the current revised manuscript. And the authors addressed the questions that I had.

We appreciate the evaluation and the positive feedback.

Reviewer #3 (Remarks to the Author):

The authors have done significant improvement and addressed most, if not all, issues. This reviewer has no more questions. Congratulations for generating a great piece of work.

We appreciate the evaluation and the positive feedback.

Reviewer #4 (Remarks to the Author):

In this revised manuscript, the authors provided additional results and descriptions to examine the nanoparticles. New experimental data support the project design and conclusions.

We appreciate the evaluation and the positive feedback.